# A DNA methylation atlas of normal human cell types

Netanel Loyfer[1,22], Judith Magenheim[2,22], Ayelet Peretz[2,22], Gordon Cann[3], Joerg Bredno[3], Agnes Klochendler[2], Ilana Fox-Fisher[2], Sapir Shabi-Porat[1], Merav Hecht[2], Tsuria Pelet[2], Joshua Moss[2,4], Zeina Drawshy[2], Hamed Amini[3], Patriss Moradi[3], Sudharani Nagaraju[3], Dvora Bauman[5], David Shveiky[5], Shay Porat[5], Uri Dior[5], Gurion Rivkin[6], Omer Or[6], Nir Hirshoren[7], Einat Carmon[8,20], Alon Pikarsky[9], Abed Khalaileh[8], Gideon Zamir[8], Ronit Grinbaum[8], Machmud Abu Gazala[8], Ido Mizrahi[8], Noam Shussman[8], Amit Korach[10], Ori Wald[10], Uzi Izhar[10], Eldad Erez[10], Vladimir Yutkin[11], Yaacov Samet[12], Devorah Rotnemer Golinkin[13], Kirsty L. Spalding[14], Henrik Druid[15,16], Peter Arner[17], A. M. James Shapiro[18], Markus Grompe[19], Alex Aravanis[3,21], Oliver Venn[3], Arash Jamshidi[3], Ruth Shemer[2], Yuval Dor[2✉], Benjamin Glaser[13✉] & Tommy Kaplan[1,2✉]

DNA methylation is a fundamental epigenetic mark that governs gene expression and chromatin organization, thus providing a window into cellular identity and developmental processes[1]. Current datasets typically include only a fraction of methylation sites and are often based either on cell lines that underwent massive changes in culture or on tissues containing unspecified mixtures of cells[2–5]. Here we describe a human methylome atlas, based on deep whole-genome bisulfite sequencing, allowing fragment-level analysis across thousands of unique markers for 39 cell types sorted from 205 healthy tissue samples. Replicates of the same cell type are more than 99.5% identical, demonstrating the robustness of cell identity programmes to environmental perturbation. Unsupervised clustering of the atlas recapitulates key elements of tissue ontogeny and identifies methylation patterns retained since embryonic development. Loci uniquely unmethylated in an individual cell type often reside in transcriptional enhancers and contain DNA binding sites for tissue-specific transcriptional regulators. Uniquely hypermethylated loci are rare and are enriched for CpG islands, Polycomb targets and CTCF binding sites, suggesting a new role in shaping cell-type-specific chromatin looping. The atlas provides an essential resource for study of gene regulation and disease-associated genetic variants, and a wealth of potential tissue-specific biomarkers for use in liquid biopsies.

Understanding how the same DNA sequence is interpreted differently in different cell types is a fundamental challenge of biology. Gene expression, DNA accessibility and chromatin packaging are well-established essential determinants of cellular phenotype. Underneath these lies DNA methylation, a stable epigenetic mark that underpins the lifelong maintenance of cellular identity.

Available human DNA methylation datasets suffer from major limitations. Multiple studies that have characterized methylomes of embryonic development, differentiation, cancer or other settings[6–9] have relied on the Illumina BeadChip platforms, which are limited to a predefined subset of 450,000 or 860,000 CpG methylation sites, representing just 3% of around 30 million CpG sites in the human genome[10]. In addition, by measuring each CpG site independently, such assays overlook coordinated patterns of DNA methylation occurring in blocks, the critical functional units of DNA methylation[11,12].

[1]School of Computer Science and Engineering, The Hebrew University of Jerusalem, Jerusalem, Israel. [2]Department of Developmental Biology and Cancer Research, Institute for Medical Research Israel-Canada, Hadassah Medical Center and Faculty of Medicine, Hebrew University of Jerusalem, Jerusalem, Israel. [3]GRAIL, Inc., Menlo Park, CA, USA. [4]Sharett Institute of Oncology, Hadassah Hebrew University Medical Center, Jerusalem, Israel. [5]Department of Obstetrics and Gynecology, Hadassah Medical Center and Faculty of Medicine, Hebrew University of Jerusalem, Jerusalem, Israel. [6]Department of Orthopedics, Hadassah Medical Center and Faculty of Medicine, Hebrew University of Jerusalem, Jerusalem, Israel. [7]Department of Otolaryngology, Hadassah Medical Center and Faculty of Medicine, Hebrew University of Jerusalem, Jerusalem, Israel. [8]Department of General Surgery, Hadassah Medical Center and Faculty of Medicine, Hebrew University of Jerusalem, Jerusalem, Israel. [9]Surgery Division, Hadassah Medical Center and Faculty of Medicine, Hebrew University of Jerusalem, Jerusalem, Israel. [10]Department of Cardiothoracic Surgery, Hadassah Medical Center and Faculty of Medicine, Hebrew University of Jerusalem, Jerusalem, Israel. [11]Department of Urology, Hadassah Medical Center and Faculty of Medicine, Hebrew University of Jerusalem, Jerusalem, Israel. [12]Department of Vascular Surgery, Shaare Zedek Medical Center, Jerusalem, Israel. [13]Department of Endocrinology and Metabolism, Hadassah Medical Center and Faculty of Medicine, Hebrew University of Jerusalem, Jerusalem, Israel. [14]Department of Cell and Molecular Biology, Karolinska Institutet, Stockholm, Sweden. [15]Department of Oncology-Pathology, Karolinska Institutet, Stockholm, Sweden. [16]Department of Forensic Medicine, The National Board of Forensic Medicine, Stockholm, Sweden. [17]Department of Medicine (H7) and Karolinska University Hospital, Karolinska Institutet, Stockholm, Sweden. [18]Department of Surgery and the Clinical Islet Transplant Program, University of Alberta, Edmonton, Alberta, Canada. [19]Papé Family Pediatric Research Institute, Oregon Health & Science University, Portland, OR, USA. [20]Present address: Department of Surgery, Samson Assuta Ashdod University Hospital, Ashdod, Israel. [21]Present address: Illumina, Inc., San Diego, CA, USA. [22]These authors contributed equally: Netanel Loyfer, Judith Magenheim, Ayelet Peretz. ✉e-mail: yuvald@ekmd.huji.ac.il; ben.glaser@mail.huji.ac.il; tommy.kaplan@mail.huji.ac.il

Most DNA methylation analyses interrogated primarily bulk tissue, thus precluding the study of minority cell types such as tissue-resident immune cells, fibroblasts or endothelial cells, whereas others analysed cultured cells, which may contain nonphysiological methylation patterns introduced in vitro[13]. As a partial solution, recent studies used single-cell RNA sequencing data from whole tissues to identify marker genes expressed in specific cell types, then identified specific CpGs whose methylation is anticorrelated with expression. These could be used on array-based methylomes to deconvolute bulk tissue and assess cell type composition or sample purity[14,15], but might be insufficiently accurate for identification of rare cellular contributions in liquid biopsies. Some studies of the human methylome did analyse isolated primary cells using whole-genome bisulfite sequencing (WGBS), but their scope was limited[2,4,5].

To overcome these limitations and to accurately characterize the human cell methylome, we performed deep genome-wide sequencing with paired-end, 150 base pair (bp)-long reads at an average sequencing depth of 30× (6.62× or greater) on fluorescent activated cell sorter (FACS)-purified populations of 39 human cell type groups obtained from freshly dissociated adult healthy tissues. We coalesced methylation patterns across the entire genome into blocks of homogeneously methylated CpG sites and used these to study variation in methylation patterns across cell types. Here we identify and characterize genomic regions that are uniquely methylated in a tissue or cell-type-specific manner, provide vignettes of their possible biological function and introduce a fragment-level deconvolution algorithm with applications such as clinical diagnosis based on circulating cell-free DNA methylation.

## Methylation atlas of human cell types

To portray genome-wide DNA methylation across a variety of cell types, we performed WGBS (150-bp-long paired-end reads to a mean depth of at least 30×) on 205 samples representing 77 primary cell types from 137 consenting donors. These were carefully sorted and mapped to the human genome (hg19, hg38). Average sample purity (that is, proportion of material from desired cell type) was over 90% as determined by flow cytometry, gene expression and DNA methylation analysis. Several samples showed lower purity (for example, colon fibroblasts 78%, smooth muscle cells (SMC) 82%, endothelial cells 86% or adipocytes 87%). Detailed descriptions of sample isolation and purity estimations, as well as sample information, are provided in Supplementary Table 1, Supplementary Figs. 1–3 and Supplementary Information.

The cell types analysed (Fig. 1) represent most major human cell types, allowing a composite view of physiological systems (for example, gastrointestinal tract, haematopoietic cells and pancreas), as well as a comparison of similar cell types in different environments (for example, tissue-resident macrophages).

The 205 methylomes show great similarities between replicates with distinctive changes between cell types in a block-like manner, as shown in Fig. 1. We sought to identify genomic regions differentially methylated in specific cell types to shed light on cell-type-specific biological processes, define cell identity and facilitate development of methylation biomarkers to identify the cellular origin of circulating cfDNA fragments[1,11,12,16–21].

We developed wgbstools, a computational machine learning suite, to represent, compress, visualize and analyse WGBS data (https://github.com/nloyfer/wgbs_tools). We segmented the genome into 7,104,162 nonoverlapping continuous blocks by identification of change points in DNA methylation patterns across multiple conditions. Each block spans highly correlated CpG sites similarly methylated in each sample but that may covary across cell types (Supplementary Information). We retained 2,783,421 methylation blocks of at least three CpGs with an average length of 544 bp (interquartile range (IQR) = 565 bp) and eight CpGs (IQR = 5 CpGs). Robust analysis of these compact genomic units is more straightforward than individual CpG

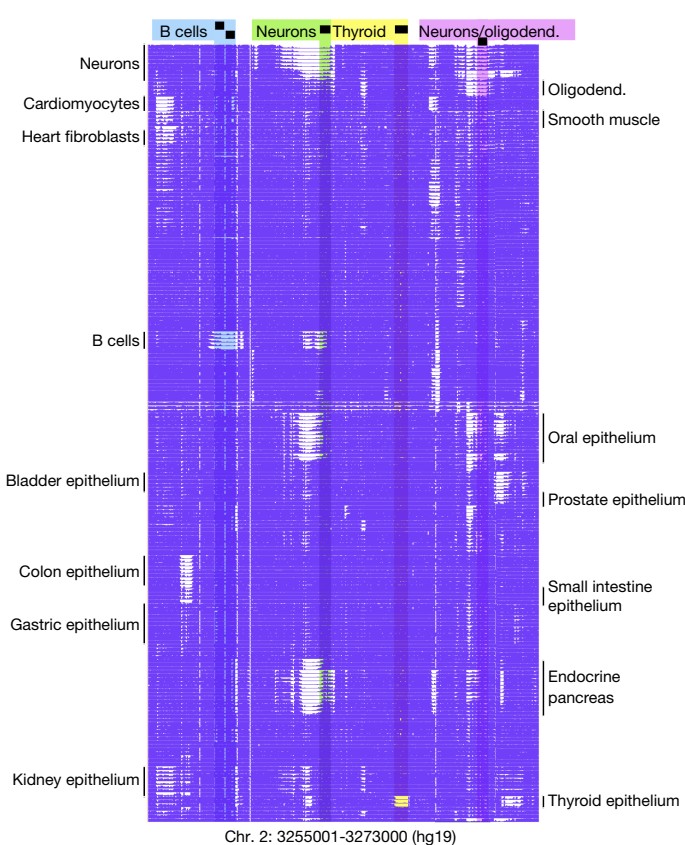

**Fig. 1 | Methylation atlas of the adult human body.** DNA methylation patterns of 205 methylomes (rows) across 344 CpG sites (columns) are demonstrated in a 18 kb region. Highlighted are regions unmethylated specifically in B cells (blue), neurons (green), thyroid epithelium (yellow) and neurons/oligodendrocytes (oliogodend.) (pink).

sites and because of the regional nature of methylation can be viewed as the biological 'atoms' of human DNA methylation[12].

## Interindividual variation in methylation

Methylation patterns were extremely robust across different individuals. For most cell types, 0.5% or less of blocks showed a difference of 50% or more across different donors compared with 4.9% among samples of different cell types (Extended Data Fig. 4). This high similarity in DNA methylation across donors is on a par with the estimated interindividual variability of genomic sequence[22]. Whereas the definition of 50% is somewhat arbitrary, other thresholds (35–50%) show a similar trend, with 0.5% or less variable blocks. Similar interindividual variation was observed in replicates obtained from different laboratories (Supplementary Table 1). Strikingly, for cell types with $n \geq 3$ biological replicates, 195 of 197 samples (99%) showed the highest similarity to another replicate (rather than to another cell type from the same donor). These results demonstrate the reproducibility of preparations but also, in agreement with previous studies[6], highlight the fundamental biological phenomenon that DNA methylation is primarily determined by cell lineage and cell-type-specific programmes rather than by genetic or environmental factors.

## Methylation records developmental history

Whereas DNA methylation patterns reflect the functional identity of a cell, they could also be used to track its developmental history.

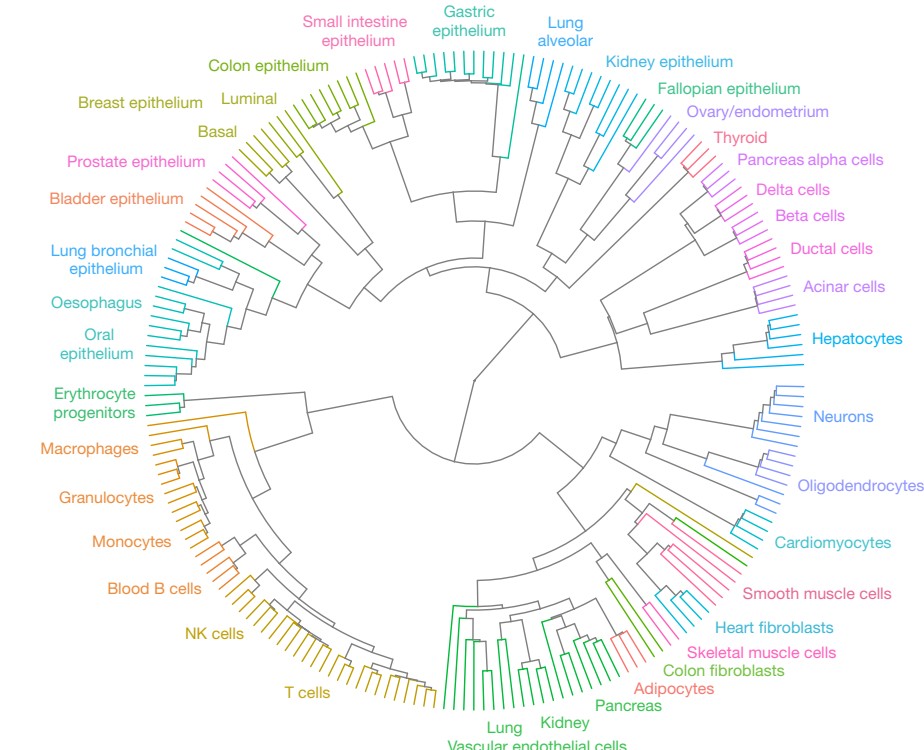

**Fig. 2 | Unsupervised agglomerative clustering reflects human developmental lineage of healthy cell types.** Cell types are indicated by edge colours.

To identify patterns shared by the progeny of early progenitors, we calculated average methylation within blocks of at least four CpGs and selected those showing the highest variability across all samples (21,000 blocks, top 1%; Supplementary Table 2). We then clustered all 205 methylomes using an unsupervised agglomerative algorithm that iteratively identifies and connects the two closest samples regardless of their labelling[23]. This analysis systematically grouped biological samples of the same cell type (Fig. 2), similar to array-based clustering of purified human blood cells[6]. This supports the reproducibility of cell isolation and suggests that three or four replicates of each normal cell type are sufficient to infer its methylation patterns for practical applications such as biomarker identification.

Strikingly, the resulting fanning diagram recapitulates key elements of lineage relationships among human tissues. For example, pancreatic islet cell types (alpha, beta and delta), which originate from the same embryonic endocrine progenitor[24], densely cluster together. Consistent with methylomes reflecting lineage rather than function, islet cells further cluster with pancreatic duct and acinar cells, and then with hepatocytes, with whom they share endodermal origins. Conversely, endoderm-derived islet cells do not cluster with ectoderm-derived neurons[25] despite common tissue-specific gene regulation and exocytosis machinery[26].

Additional examples include the clustering of gastric, small intestine and colon epithelial cells; the clustering of all blood cell types; and the clustering of multiple mesoderm-derived cell types including vascular endothelial cells, adipocytes and skeletal muscle. Interestingly, lung bronchial epithelium clustered with oesophagus and oral epithelium whereas lung alveolar epithelium clustered with intestinal epithelium, consistent with evidence of early developmental origins of the alveolar cell lineage[27].

Some methylation patterns were common to lineages that formed during early developmental stages. For example, 892 regions were unmethylated in epithelial cells derived from early endodermal derivatives and methylated in mesoderm- and ectoderm-derived cells (Methods). We suggest that these were demethylated in the endoderm germ layer, with derived cell types retaining these patterns decades later (Extended Data Fig. 5a). Because endoderm derivatives do not share common function or gene expression, this provides yet another example of methylation patterns as a stable lineage mark.

Finally, we applied the same segmentation and clustering approach to a published methylation atlas from the Roadmap Epigenomics project[4]. The algorithm did not group related cell types, and often clustered samples based on donor identity. This further emphasizes the importance of careful purification of homogeneous cell types, avoiding mixed cell populations (Extended Data Fig. 5b).

## Cell-type-specific methylation markers

We next turned to study genomic regions differentially methylated in a cell-type-specific manner. We organized the 205 samples into 39 groups of specific cell types, including blood cell types (B, T, natural killer (NK), granulocytes, monocytes and tissue-resident macrophages), breast epithelium (basal and luminal), lung epithelium (alveolar and bronchial), pancreatic endocrine (alpha, beta and delta) and exocrine (acinar and duct) cells, vascular endothelial cells from various sources, cardiomyocytes and cardiac fibroblasts and more. We also defined 12 supergroups in which related cell types were grouped, including muscle cells, gastrointestinal epithelium, pancreas and more (Supplementary Table 3).

We then focused on differentially methylated blocks comprising five or more CpGs that are unmethylated in one group of cell types but methylated in all other samples, or vice versa. Intriguingly, almost all regions (97%) were unmethylated in one cell type and methylated in all others. We then sorted these differential regions by absolute difference in methylation in target cell type versus all other samples (Methods and Supplementary Information).

The top 25 differentially unmethylated regions for each cell type comprise a human cell-type-specific methylation atlas of 1,246 markers (Fig. 3 and Supplementary Table 4). These regions are uniquely unmethylated in particular cell types (average methylation 13%) and methylated in all other samples (average methylation 91%), and can serve as sensitive biomarkers for quantification of the presence of DNA from a specific cell type in a mixture. The markers include 953 cell-type-specific

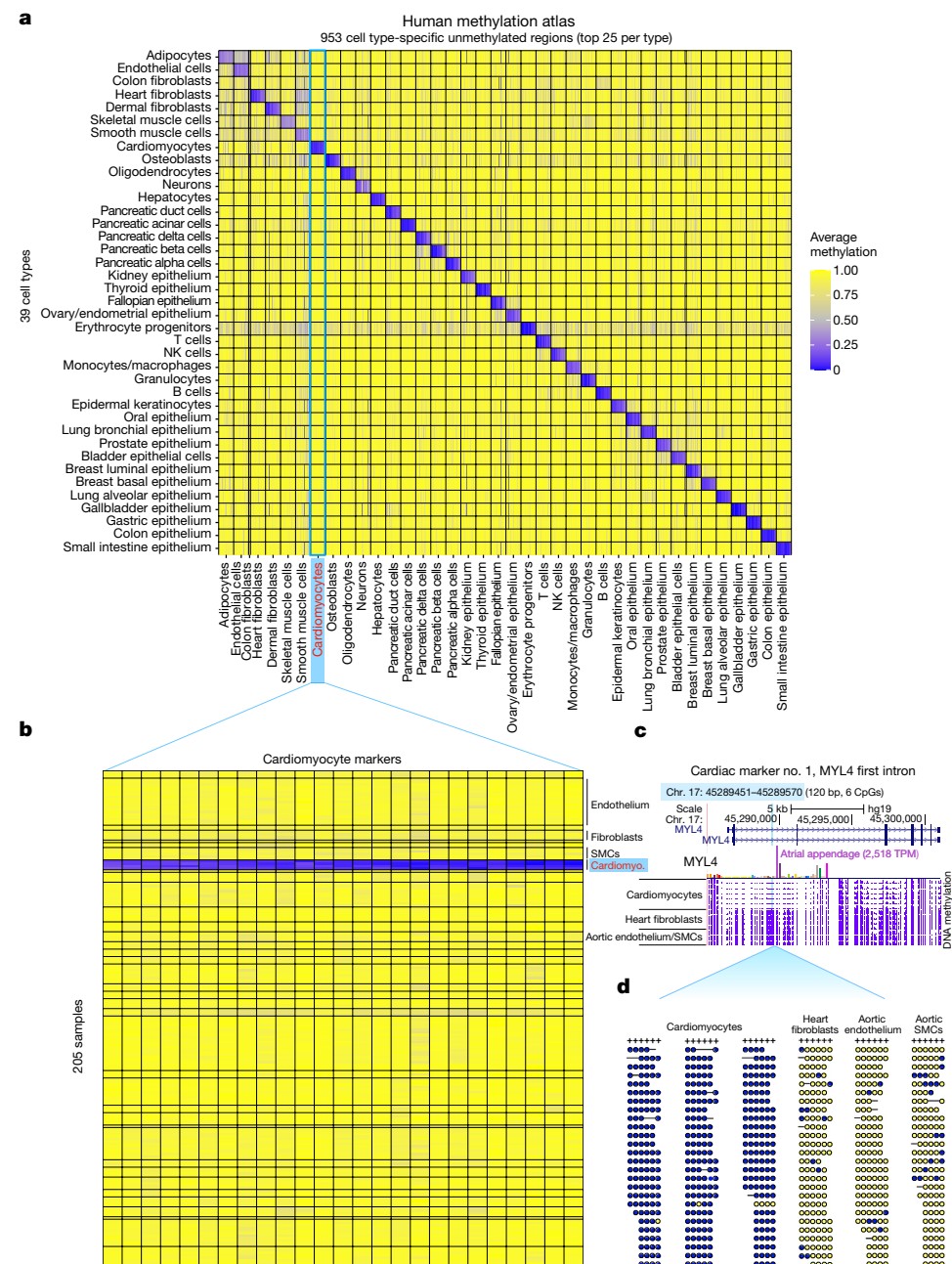

**Fig. 3 | A human methylation atlas of 205 samples across 39 cell type groups. a**, A total of 953 genomic regions unmethylated in a cell-type-specific manner. Each cell in the plot marks the average methylation of one genomic region (column) at each of 39 cell types (rows). Up to 25 regions are shown per cell type, with a mean length of 356 bp (nine CpGs) per region. **b**, The top 25 cardiomyocyte regions. For each region we plot the average methylation of each CpG site (columns) across all 205 samples in the atlas, grouped into 39 cell types as before. **c**, A locus specifically unmethylated in cardiomyocytes. This marker (highlighted in light blue) is 120 bp in length (six CpGs) and is located in

the first intron of MYL4, a heart-specific gene (transcripts per million (TPM) expression of 2,518 in atrial appendage, GTEx inset). Genomic snapshot depicts average methylation (purple tracks) across six cardiomyocyte samples, four cardiac fibroblast samples and three aorta samples (two endothelial and one SMC). **d**, Visualization of bisulfite-converted fragments from three cardiomyocyte samples, one cardiac fibroblast sample and two aorta samples (endothelium and SMC). Shown are reads mapped to chr. 17: 45289451–45289570 (hg19), with at least three covered CpGs. Yellow and blue dots denote methylated and unmethylated CpG sites, respectively.

unmethylated loci, as well an additional 293 loci that are unmethylated in few related cell types. A fragment-level analysis further shows that the vast majority of DNA fragments at these regions are unmethylated in the target cell type compared with almost none in all other cell types (Extended Data Fig. 6). The atlas has various applications, including the analysis of circulating cell-free DNA fragments[18–21,28–30]. Importantly, only about 1% of cell type-specific markers are covered by reduced representation bisulfite sequencing (RRBS), 4–8% by methyl-sequencing

hybrid capture panels and 14–24% are represented in single-CpG 450K/EPIC arrays[10], emphasizing the benefits of whole-genome sequencing for exhaustive identification of biomarkers.

## Human cell-type-specific regulatory maps

We next turned to characterize these sets of cell-type-specific differentially unmethylated regions. For this we identified the top

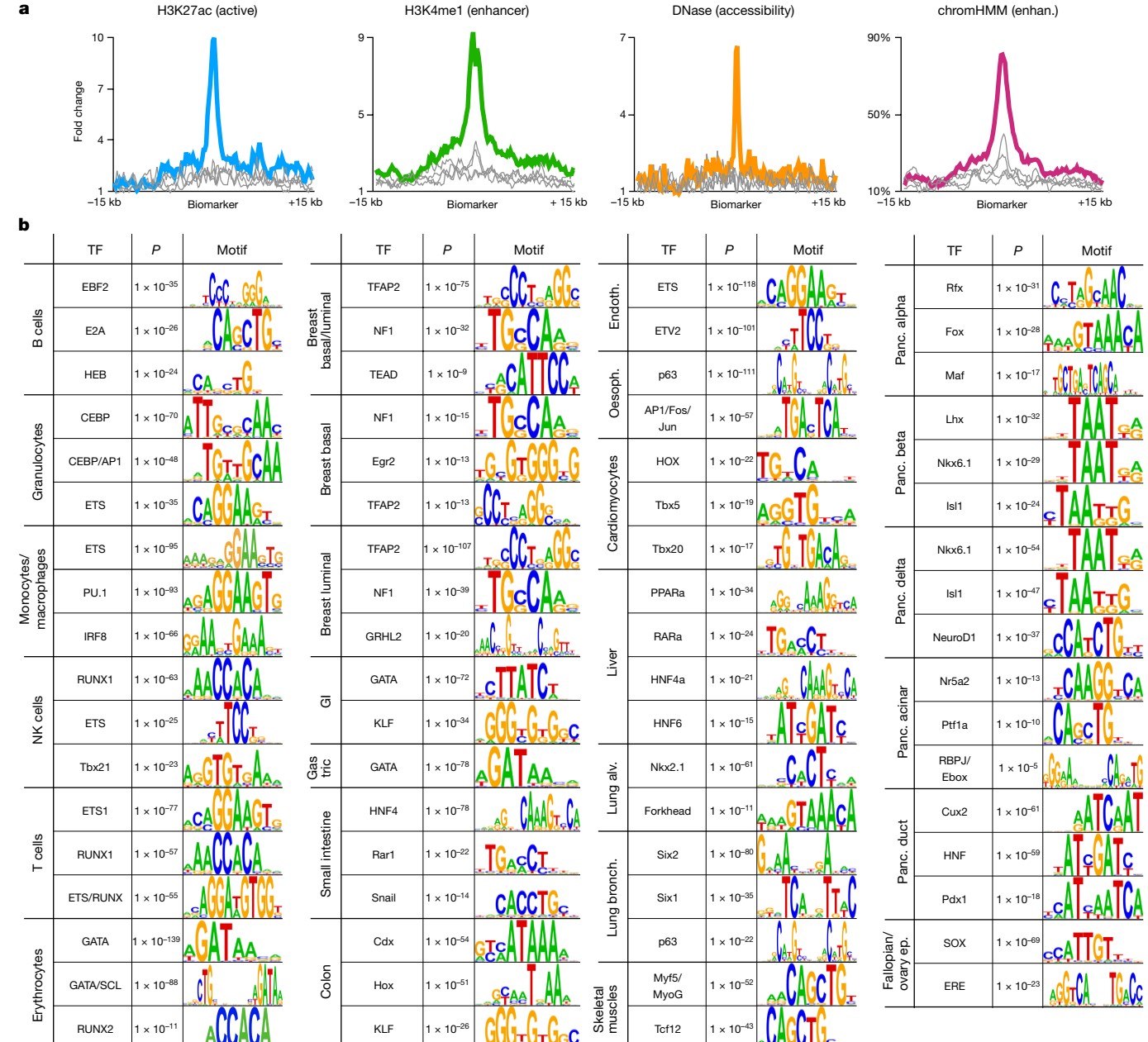

**Fig. 4 | Cell-type-specific markers as putative enhancers. a**, Average ChIP–seq signal for the active regulatory marker H3K27ac, enhancer marker H3K4me1, DNA accessibility and chromHMM enhancer annotations for the top 250 cell-type-specific unmethylated markers for monocytes/macrophages. Average signal for the top 250 markers of other blood cell types (granulocytes and B, T and NK cells) shown as grey lines, for comparison. **b**, Cell-type-specific 250 unmethylated markers for each cell type (Supplementary Table 4b) are enriched for regulatory motifs. Shown are the top TF binding site motifs, enriched among the top 1,000 differentially unmethylated regions per cell type, using HOMER motif analysis. Motifs similar to previous (more significant) hits not included. Shown are HOMER binomial $P$ values. Alv., alveolar; Bronch., bronchial; Endoth., endothelium; Ep., epithelium; Oesoph., oesophagus; Panc., pancreas.

250 unmethylated markers for each cell type (Supplementary Table 4b) and used GREAT to identify those genes adjacent to each group of markers, and to test their enrichment for various gene-set annotations[31]. Genes adjacent to loci uniquely unmethylated in a given cell type typically reflect the functional identity of that cell type. For example, genes near B cell markers were enriched for B cell morphology, differentiation, IgM levels and lymphopoiesis; NK cell markers were associated with NK cell-mediated cytotoxicity, the haematopoietic system, cytotoxicity and lymphocyte physiology; Fallopian tube markers were enriched for egg coat and perivitelline space; and cardiomyocyte markers for cardiac relaxation, systolic pressure, muscle development and hypertrophy (Supplementary Table 5).

We then analysed the DNA accessibility and chromatin packaging of cell-type-specific markers as defined by assay for transposase-accessible chromatin using sequencing (ATAC–seq), DNase I hypersensitive site sequencing (DNaseI–seq)[4,32] and histone marks indicative of active promoters and enhancers[4]. The top 250 unmethylated markers for monocytes and macrophages are highly accessible and characterized by H3K27ac and H3K4me1 in monocytes, whereas markers of other cell types show no enrichment in monocytes (Fig. 4a), with similar results for markers of other cell types (Extended Data Fig. 7). We also show strong coordinated enrichment of chromHMM enhancer annotations at cell-type-specific markers[33] (Fig. 4a). These findings are consistent with previous studies that have associated tissue-specific demethylation with gene enhancers[1,34].

To further assess the biological importance of cell-type-specific unmethylated regions, we studied their association with transcription factors (TFs) that could either affect DNA methylation or bind DNA in a cell-type-specific manner, depending on methylation and chromatin[35–38]. We identified the top 1,000 unmethylated markers per cell type (Supplementary Table 4c) and performed motif analysis using HOMER[39] to calculate the enrichment of known TF binding motifs (Supplementary Table 6a). For most cell types the top motifs included master regulators and key TFs (Fig. 4b). For example, B cells are enriched for Ebf2/HEB/E2A, granulocytes for CEBP/AP1/ETS and T cells for ETS/RUNX. This association between cell-type-specific unmethylated regions and TF binding motifs can identify new gene regulatory circuits and expose distal enhancers active in specific cell types.

We aimed to identify the target genes of putative enhancers marked by cell-type-specific demethylation. Top markers frequently fall within intronic regions and are likely to regulate these genes (for example, glucagon in pancreatic alpha cells, NPPA and MYL4 in cardiomyocytes and MBP in oligodendrocytes; Supplementary Table 7), or proximally to probable targets (for example, a beta cell marker 5 kb from the insulin gene). Other markers are further apart from their target genes. We devised a computational algorithm to identify genes in the proximity of cell-type-specific markers showing increased gene expression levels under matching conditions (Methods). This highlighted hallmark genes for many cell types and suggested putative targets for many of the top 25 unmethylated markers for each cell type. For example, hepatocyte markers were associated with APOE, APOC1, APOC2 and the glucagon receptor. Similarly, cardiomyocyte markers were associated with NPPA, NPPB and myosin genes; and pancreatic islet markers with insulin and glucagon genes (Supplementary Table 7). These findings further support the principle that loci specifically unmethylated in a given cell type are probably enhancers positively regulating genes expressed in this cell type, often controlling adjacent genes. We note, however, that genes adjacent to a locus specifically unmethylated in a given cell type are often broadly expressed beyond this cell type (Discussion).

To generate a catalogue of putative regulatory regions in each cell type we applied a fragment-level analysis across all samples from each cell type, independently of other cell types. We scanned the entire genome and identified genomic regions in which at least 85% of DNA fragments with at least four CpGs are unmethylated (Methods). This identified a set of unmethylated genomic regions in each of the 39 cell type groups analysed, including 36,111 regions on average (Supplementary Dataset 1). These regions were then annotated for genomic features, showing that 56% on average overlapped CpG islands, 46% were near promoter regions and 44% overlapped CTCF binding sites, thus highlighting the regulatory and structural roles of unmethylated loci. When available, we crossed these regions with chromatin immunoprecipitation sequencing (ChIP–seq) peaks from ENCODE[5] and Roadmap Epigenomics[4] under matching conditions, including H3K4me3, H3K27ac, H3K4me1, H3K27me3, CTCF and ATAC–seq, and generated a cell-type-specific catalogue of putative enhancer regions comprising unmethylated regions that overlap H3K27ac, but not H3K4me3, peaks (Supplementary Dataset 2). Motif analysis of these regions identified key TFs in each cell type, similar to those shown in Fig. 4 (Supplementary Table 6b,c).

## Cell-type-specific hypermethylated loci

We studied those genomic regions methylated in one cell type but unmethylated elsewhere in the human body. These are enriched for CpG islands (38% of methylated regions compared with 1.7–2.7% of cell-type-specific unmethylated regions), and are marked by H3K27me3 and Polycomb in other cell types (Fig. 5a–c), as previously reported for cancer and developmental processes[40,41]. These cell-type-specific hypermethylated regions were generally less significant for motif enrichment (compared with uniquely unmethylated regions).

Intriguingly, only around 3% of the total set of cell-type-specific differentially methylated regions are hypermethylated.

After pooling all cell-type-specific hypermethylated regions, we identified strong enrichment for target sequences of the chromatin regulator CTCF ($P \leq 1 \times 10^{-18}$; Fig. 5d). This suggests that DNA methylation of CTCF binding sites could act as a tissue-specific regulatory switch to modulate its binding, potentially affecting tissue-specific three-dimensional genomic organization[35,42,43]. To test this idea we compared patterns of DNA methylation at CTCF sites with genome-wide CTCF protein binding in specific tissues. Figure 5e shows the methylation pattern and published in vivo CTCF occupancy at one locus, which is methylated specifically in the colon and intestine. Consistent with DNA methylation preventing CTCF binding, ChIP data show selective absence of CTCF binding at this locus in the colon. In addition, loci methylated in specific cell types were enriched for targets of the transcriptional repressor of neural genes, RE1-silencing TF/neuron-restrictive silencer factor (REST/NRSF) ($P \leq 1 \times 10^{-24}$), and this was seen most prominently in the methylome of pancreatic islet cells (Fig. 5f). Whereas DNA methylation has not been shown to affect the binding or activity of REST, this finding raises the intriguing possibility that methylation of REST targets in islets could permit endocrine differentiation independently of REST repression.

## Fragment-level methylome deconvolution

Last, we developed a computational fragment-level deconvolution algorithm for DNA methylation sequencing data and used the top 25 markers defined for each cell type (a total of 1,246 markers) to study methylomes obtained from composite tissue samples and cfDNA. Briefly, we generated an atlas in which the percentage of unmethylated fragments is computed for every marker (row) in each cell type (column). A non-negative least-squares (NNLS) algorithm is then used to fit an input sample and estimate its relative contributions (Supplementary Information).

To estimate the accuracy of our fragment-level approach, we used in silico mixtures of sequenced reads. For each cell type we applied a leave-one-out approach to mix one held-out sample in leukocyte reads, then used the deconvolution algorithm to infer cellular composition in the mixture. We repeated this process at concentrations varying from 0 to 10%. As shown in Fig. 6a, we found that the 1,246 markers (top 25 per cell type) allowed accurate detection of DNA from a given source at around 0.1% resolution, an improvement of nearly one order of magnitude in comparison with array-based approaches[28]. Four-way in silico mixes, in which endothelial and hepatocyte methylomes were also included to realistically mimic cfDNA composition, yielded similar results (Extended Data Fig. 8).

We then estimated the cellular composition of leukocytes and cfDNA using WGBS data from 23 healthy donors; 99.5% of leukocyte-derived DNA was attributed to granulocytes, monocytes, macrophages and NK, T and B cells, consistent with typical blood counts (Fig. 6b and Supplementary Table 8). The cfDNA of healthy subjects was mostly derived from leucocytes: granulocytes (29.7%), monocytes/macrophages (20%) and lymphocytes (3%). Solid tissues contributing to cfDNA included vascular endothelial cells (6%) and hepatocytes (3.1%) (Fig. 6c), consistent with previous results[28]. The current atlas also shows a significant contribution of megakaryocytes (31%) and erythrocyte progenitor (prog.) cells (5%) to cfDNA, which were not observed in previous studies that used reference methylomes of a more limited scope.

## Endothelial cfDNA in patients with COVID-19

Analysis based on DNA methylation patterns offers an opportunity to identify the tissue origins of cfDNA. COVID-19 inflicts damage to multiple tissues, some of which have no biomarkers. We used the atlas

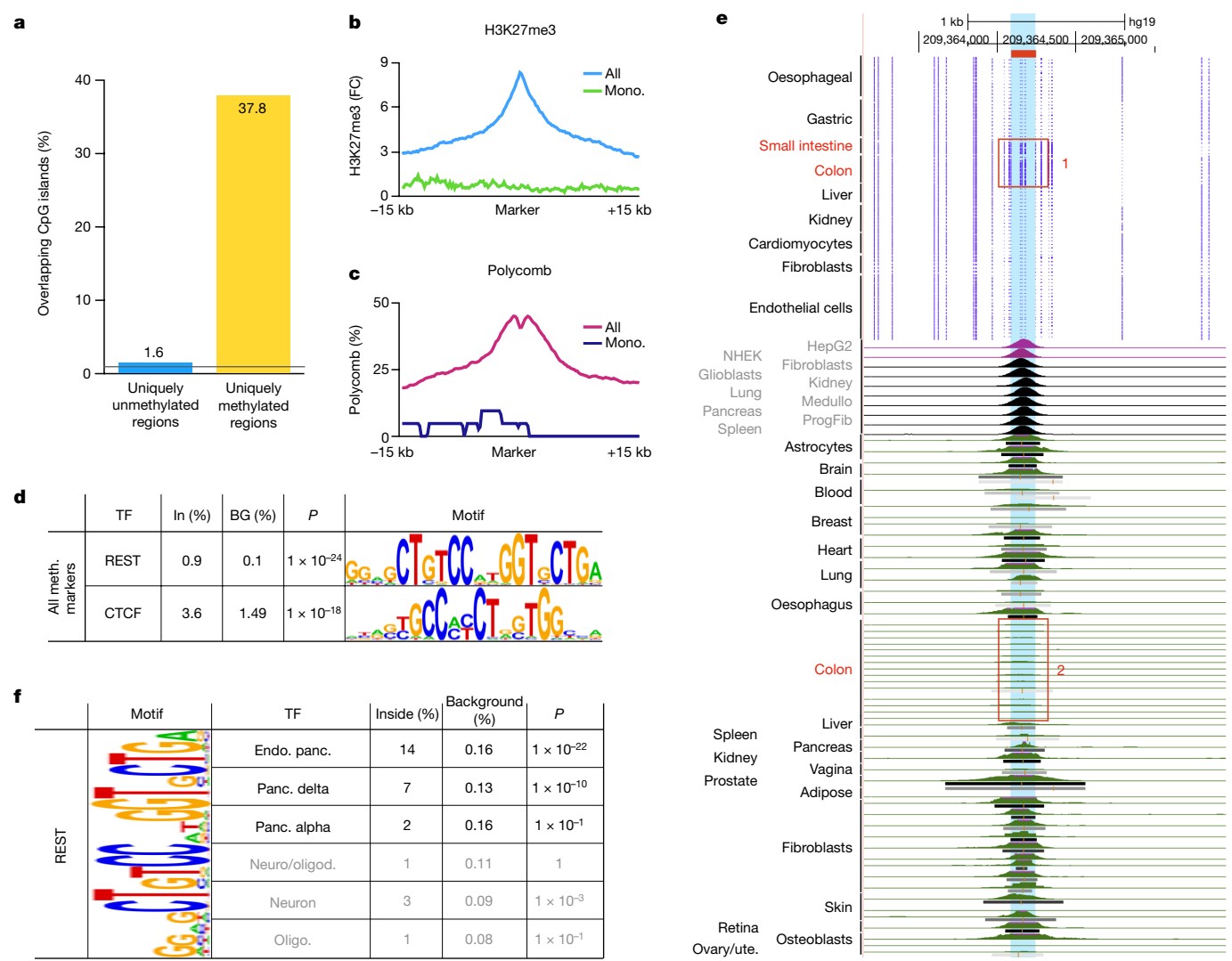

**a** Overlapping CpG islands (%): Uniquely unmethylated regions 1.6; Uniquely methylated regions 37.8

**b** H3K27me3 — All, Mono. −15 kb, Marker, +15 kb

**c** Polycomb (%) — All, Mono. −15 kb, Marker, +15 kb

**d**

| | TF | In (%) | BG (%) | $P$ | Motif |
|---|---|---|---|---|---|
| All meth. markers | REST | 0.9 | 0.1 | $1 \times 10^{-24}$ | |
| | CTCF | 3.6 | 1.49 | $1 \times 10^{-18}$ | |

**f**

| Motif | TF | Inside (%) | Background (%) | $P$ |
|---|---|---|---|---|
| REST | Endo. panc. | 14 | 0.16 | $1 \times 10^{-22}$ |
| | Panc. delta | 7 | 0.13 | $1 \times 10^{-10}$ |
| | Panc. alpha | 2 | 0.16 | $1 \times 10^{-1}$ |
| | Neuro/oligod. | 1 | 0.11 | 1 |
| | Neuron | 3 | 0.09 | $1 \times 10^{-3}$ |
| | Oligo. | 1 | 0.08 | $1 \times 10^{-1}$ |

**Fig. 5 | Cell-type-specific hypermethylated regions are enriched for CpG islands, Polycomb targets and CTCF and REST/NSRF. a,** Thirty-eight per cent of the top cell type-specific hypermethylated markers (1,363 of 3,613, binomial $P < 1 \times 10^{-100}$) overap CpG islands. By comparison, 1.6% of cell-type-specific hypomethylated regions (189 of 11,714) overlap CpG islands, comprising less than 0.9% of the genome (black line). **b,** These regions are typically enriched for H3K27me3 in other cell types. Shown are average H3K27me3 signals in monocytes and macrophages near all cell-type-specific hypermethylated regions (blue) or near monocyte/macrophage-specific hypermethylated regions (mono; green). **c,** Similar plots for Polycomb annotations in monocytes and macrophages (chromHMM), for all or monocyte/macrophage-specific markers. **d,** Motif analysis of cell-type-specific hypermethylated regions (top 100 per cell type) identifies known CTCF and REST/NSRF motifs. HOMER binomial $P$ values are shown. **e,** Analysis of ChIP–seq data for one such site (chr. 1: 209364093–209364250, highlighted in blue, hg19), specifically methylated in the small intestine and colon epithelium (red box 1) and unmethylated elsewhere. As shown below, this site is bound in multiple cell types and tissues but is mostly unbound in stomach and colon epithelium in vivo (red box 2). **f,** REST/NSRF motif is present within 14% of the top 100 cell type-specific hypermethylated regions in the endocrine pancreas, 7% of top delta cell markers and 2% of top alpha cell markers, compared with approximately 0.1% in background sequences, in accordance with REST target expression in the endocrine pancreas. HOMER binomial $P$ values are shown. Alv., alveolar; bronch., bronchial; Endo. panc., endocrine pancreas; Ep., epithelium; Oesoph., oesophagus; Oligo, oligodendrocytes; Panc., pancreas; Ute., uterus.

to deconvolve shallow WGBS data from 52 patients hospitalized owing to COVID-19 (ref. [44]). We identified excessive cell-free DNA fragments from granulocytes, erythrocyte progenitors, lung and liver, consistent with published analysis of these samples (Supplementary Information). Strikingly, we also identified a significant contribution of vascular endothelial cells to the cfDNA of these patients, which could not be detected in the published analysis in the absence of an endothelial cell methylome reference (Fig. 6d). Interestingly, the concentration of endothelial cell-derived cfDNA was higher in patients with severe disease (WHO score ≥7) compared with those with milder disease (WHO score ≤6; $P \le 6 \times 10^{-5}$, Mann–Whitney). These results suggest that vascular endothelial cell death plays a substantial role in the pathogenesis

of COVID-19, potentially related to coagulopathy, and highlight the benefit of using a comprehensive cell-type-specific atlas for cfDNA methylome analysis.

## Cell type deconvolution of composite tissues

Finally, we analysed whole-genome methylomes from ENCODE[5] and the Roadmap Epigenomics atlas[4] using our atlas (based on 25 markers per cell type). Deconvolution of some methylomes showed a homogenous composition as intended—for example, 97–99% T cell DNA in Roadmap T cell samples (Supplementary Table 9). However, analysis of

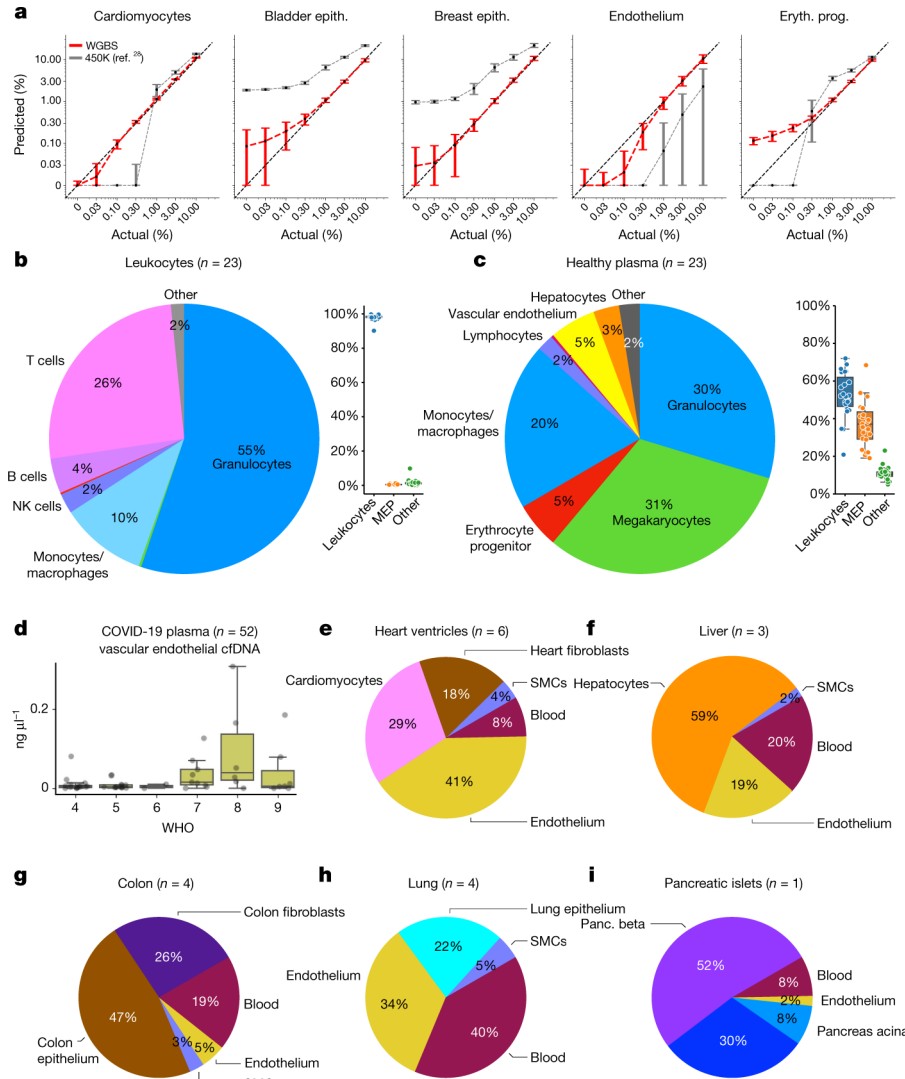

**Fig. 6 | Fragment-level deconvolution using cell type-specific biomarkers.** **a**, Cell-type-specific markers achieved less than 0.1% resolution. In silico simulations for five cell types, in which held-out samples were computationally mixed within leukocytes then analysed using 1,246 atlas markers plus 25 additional megakaryocyte markers (red) or an array-based deconvolution of these mixes[28] (grey). Box plots show average contribution in ten simulations, with error bars representing 1 s.d. **b,c**, Cell type composition in leukocytes (**b**) and plasma samples (**c**) from healthy donors. Box plots show overall proportions of leukocytes, megakaryocytes and erythroblasts (MEP) and other cell types. **d**, Analysis of low-coverage plasma samples from 52 patients with SARS-CoV-2

(ref. [44]) identified endothelial-derived cfDNA in patients with WHO ordinal scale seven or higher (requiring admittance to intensive care unit). **e–i**, Fragment-level deconvolution of Roadmap/ENCODE samples[4,5] showing cell-type-specific contributions. **e**, Heart ventricle samples contained a mixture of cardiomyocytes, endothelial cells, fibroblasts and blood. **f**, Liver samples contained around 60% of hepatocyte DNA, plus blood and endothelial cells. **g**, Colon samples contained approximately 50% epithelium, plus fibroblasts and blood. **h**, Lung samples contained less than 30% of lung epithelial cells. **i**, Pancreatic islet samples contained beta, alpha, duct and acinar cells. Box plots denote median and IQR, with whiskers 1.5× IQR.

other samples showed a highly heterogeneous composition, as previously reported based on array-based bulk tissue deconvolution algorithms such as EpiDISH and EpiScore[14,15,45]. For example, heart ventricle samples comprised 29% cardiomyocytes, 41% endothelial cells and 18% cardiac fibroblasts (Fig. 6e); liver methylomes comprised around 60% hepatocytes, 21% blood and 20% endothelial cells; and colon methylomes comprised about 50% colon epithelium, 26% colon fibroblasts and 19% blood. Most strikingly, Roadmap lung samples were dominated by blood (40%), endothelium (34%) and smooth muscle (5%), with only 22% of DNA derived from lung epithelial cells (Fig. 6f–i and Supplementary Table 9). Importantly, a similar deconvolution of the 205 samples presented here yielded an average contribution of 94% for the expected cell type for each sample (median of 95%, Supplementary Table 10), or of 91% (median of 92%) in a more stringent leave-one-out cross-validation analysis (Supplementary Table 11), highlighting the purity of collected samples.

Naturally, fragment-level analysis is limited to cell types for which whole-genome sequencing data are available, and some cell types can be analysed only by array-based algorithms[15,28]. Nonetheless, the markers and algorithm presented here allow analysis of composite bulk tissue and plasma samples, across multiple cell types and with high accuracy.

## Discussion

The comprehensive atlas of human cell type methylomes described here sheds light on principles of DNA methylation and provides a valuable resource for multiple lines of investigation, as well as translational applications.

Our analysis used whole-genome sequencing data to show that methylation patterns are strikingly similar among healthy replicates

of the same cell type from different individuals. The similarity between individuals reflects the robustness of cell differentiation and maintenance circuits, at least as far as healthy tissues are concerned. Pathologies involving destabilization of the epigenome obviously disrupt these circuits, resulting in a larger variety of methylation patterns among cells descended from a specific normal cell type. We predict that, even in cancers (of the same primary anatomic site and histologic type), comparative methylome analysis of purified epithelial cells, performed at the level of methylation blocks, will show a smaller interindividual variation than typically assumed.

As the atlas demonstrates, each cell type has a set of genomic regions that are uniquely unmethylated in that cell type compared with others, as well as additional genomic regions that share methylation patterns with related cell types. Using unsupervised clustering of cell-type-specific methylomes, we found that cell types were clustered in ways that reflected their developmental origins rather than expression patterns. This offers a fascinating view of DNA methylation as a record of the methylomes of progenitor cells, retained in the genome through dramatic developmental transitions and decades of life thereafter. We propose that comparative methylome analysis will allow reconstruction of parts of the methylomes of fetal structures or cell types, similarly to the reconstruction of last common ancestors in evolutionary biology.

The vast majority of cell-type-specific differentially methylated regions were specifically demethylated in one cell type. The chromatin of these regions is typically highly accessible and bears histone marks associated with active gene regulation, as found in enhancers and promoters. Moreover, these loci are enriched for TF binding site motifs that operate in that cell type. We devised an integrated approach that, based on distance and gene expression profiles, allowed us to highlight potential target genes for these putative enhancer regions. Many enhancer regions were associated with nearby genes that are broadly expressed, potentially reflecting gene regulation by multiple tissue-specific enhancers. Our findings are consistent with previous studies that showed tissue-specific hypomethylation occurring at gene enhancers[35–37]. Our data-driven approach for marker identification is complementary to recent gene-centric approaches[14,15] that use tissue-specific single-cell RNA sequencing data to define marker genes and identify neighbouring CpGs specifically unmethylated in target cell types. Finally, we devised a fragment-level genomic analysis to identify tens of thousands of unmethylated regions, per cell type, which were annotated with genomic features, DNA accessibility, chromatin marks and TF binding motifs to produce a cell type-specific catalogue of putative enhancers. Further analysis of this atlas will show and validate the complete set of human enhancers in each cell type.

Conversely, we identified genomic regions specifically methylated in one or two cell types, representing around 3% of cell-type-specific differentially methylated regions. These are often located in CpG islands and characterized by H3K27me3 and Polycomb binding in tissues where the locus is not methylated[40,41]. This epigenetic repressive switching was previously described in cancer and during early development[41,46], but its role during differentiation of specific cell types remains unclear. These regions are enriched for CTCF binding sites, suggesting a role for DNA methylation in attenuating the binding of CTCF and thus modulation of the cell-type-specific, three-dimensional organization of neighbouring DNA[35,36,47].

For DNA methylation sequencing data, the atlas described here is, to our knowledge, the most comprehensive compendium to date. We identified more than one thousand cell-type-unique DNA methylation regions that could serve as accurate and specific biomarkers for fragment-level analysis and identification of cell death events by monitoring of cfDNA. Notably, most of these marker regions are not covered by 450K/EPIC BeadChip DNA methylation arrays, and were not previously appreciated. To allow interpretation of array data, we offer alternative sets of cell-type-specific markers limited to CpG sites included in BeadChip 450K arrays. Similarly, we identified cell-type-specific markers in regions targeted by both RRBS and hybrid capture panels (Extended Data Fig. 9 and Supplementary Tables 12–17). As shown in Extended Data Fig. 10, the array-adapted atlas allows high-resolution interpretation of array methylomes of pancreatic islet, lung and breast biopsies, highlighting the presence of cell types not previously profiled[48–50].

Many cell types are missing from the atlas, typically because of limited availability of material. Examples include osteoblasts, cholangiocytes, cells of the adrenal gland, urethral epithelium and haematopoietic stem cells. Additionally, we did not separate many subpopulations of interest—for example, different types of neurons or lymphocytes. The atlas is viewed as a living, publicly available database to be updated in the future. The resolution of the atlas yields a quantitative understanding of composite tissues and allows one to identify missing methylomes of additional cell types yet to be characterized. We also acknowledge that the purity of the sorted cell populations varies, owing to variation in the quality of antibodies used for FACS and the extent to which they allow separation of cell types. Nonetheless, even the least pure cell types in the atlas (for example, some preparations of vascular endothelial cells, fibroblasts, SMC and adipocytes showing 70–80% purity), when averaged over replicates, are useful for identification of differentially methylated regions and for inference of cell composition in mixtures.

In summary, we present a comprehensive methylation atlas of primary human cell types along with an extensive set of cell-type-specific markers and computation tools for fragment-level analysis of mixed cell type samples. These complement the plethora of array-based methylomes and deconvolution tools available for the analysis of array data. Together, the data shed light on the roles of DNA methylation in cellular biology and gene regulation and facilitate the identification of enhancers active in each cell type. Perhaps the most promising utility of our atlas is the potential for fragment-level deconvolution of mixed cell type samples, allowing sensitive identification of the tissue of origin of cfDNA in plasma of individuals with cancer and other diseases[18–21,28–30].

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

## Methods

### Human tissue samples

Human tissues were obtained from various sources, as detailed in Supplementary Table 1. The majority (148) of the 205 samples analysed were sorted from tissue remnants obtained at the time of routine, clinically indicated surgical procedures at the Hadassah Medical Center. In all cases, normal tissue distant from any known pathology was used. Surgeons and/or pathologists were consulted before removal of tissue to confirm that its removal would not compromise the final pathologic diagnosis in any way. For example, in patients undergoing right colectomy for carcinoma of the caecum, the most distal part of the ascending colon and most proximal part of the terminal ileum were obtained for cell isolation. Normal bone marrow was obtained at the time of joint replacement in patients with no known haematologic pathology. The patient population included 135 individuals ($n = 60$ males, $n = 74$ females) aged 3–83 years. The majority of donors were White. Approval for collection of normal tissue remnants was provided by the Institutional Review Board (IRB, Helsinki Committee), Hadassah Medical Center, Jerusalem, Israel. Written informed consent was obtained from each donor or legal guardian before surgery.

As described in Supplementary Table 1, some cells and tissues were obtained through collaborative arrangements: pancreatic exocrine and liver samples (cadaveric organ donors, $n = 5$) from M. Grompe, Oregon Health & Science University; adipocytes (subcutaneous adipocytes at time of cosmetic surgery following weight loss, $n = 3$), oligodendrocytes and neurons (brain autopsies, $n = 14$) from K. L. Spalding and H. Druid, Karolinska Institute, Stockholm; and research-grade cadaveric pancreatic islets from J. Shapiro, University of Alberta ($n = 16$). In all cases, tissues were obtained and transferred in compliance with local laws and after the approval of the local ethics committee on human experimentation. Sixteen cell types were obtained from commercial sources, including 15 from Lonza and one from Sigma-Aldrich. Three pancreatic islet preparations were obtained from the Integrated Islet Distribution Program (https://iidp.coh.org).

### Tissue dissociation and FACS sorting of purified cell populations

Fresh tissue obtained at the time of surgery was trimmed to remove extraneous tissue. Cells were dispersed using enzyme-based protocols optimized for each tissue type. The resulting single-cell suspension was incubated with the relevant antibodies and FACS sorted to obtain the desired cell type (Extended Data Fig. 2 and Supplementary Information).

Purity of live sorted cells was determined by messenger RNA analysis for key known cell-type-specific genes, whereas the purity of cells fixed before sorting was determined using previously validated cell-type-specific methylation signals (Extended Data Fig. 2c and Supplementary Information). DNA was extracted using the DNeasy Blood and Tissue kit (no. 69504, Qiagen) according to the manufacturer's instructions, and stored at −20 °C for bisulfite conversion and whole-genome sequencing.

### WGBS

Up to 75 ng of sheared genomic DNA was subjected to bisulfite conversion using the EZ-96 DNA Methylation Kit (Zymo Research), with liquid handling on a MicroLab STAR (Hamilton). Dual-indexed sequencing libraries were prepared using Accel-NGS Methyl-Seq DNA library preparation kits (Swift BioSciences) and custom liquid handling scripts executed on the Hamilton MicroLab STAR. Libraries were quantified using KAPA Library Quantification Kits for Illumina Platforms (Kapa Biosystems). Four uniquely dual-indexed libraries, along with the 10% PhiX v.3 library (Illumina), were pooled and clustered on an Illumina NovaSeq 6000 S2 flow cell followed by 150 bp, paired-end sequencing. Total read count and average sequencing depth (in read pairs), as well as percentage of CpGs, per sample, at 1× and 10×, are detailed in Supplementary Table 1. Also listed are average methylation levels, per sample, at CpG, nonCpG and CC dinucleotides. Intriguingly, sorted neuron samples showed higher CpA methylation (approximately 10%) compared with other samples (approximately 1%).

### WGBS computational processing

Paired-end FASTQ files were mapped to the human (hg19, hg38), lambda, pUC19 and viral genomes using bwa-meth (v.0.2.0)[51] then converted to BAM files using SAMtools (v.1.9)[52]. Duplicated reads were marked by Sambamba (v.0.6.5) with parameters '-l1 -t16 --sort-buffer-size 16000 --overflow-list-size 10000000' (ref. [53]). Reads with low mapping quality, duplicated or not mapped in a proper pair were excluded using SAMtools view with parameters '-F 1796 -q 10'. Reads were stripped from nonCpG nucleotides and converted to PAT files using wgbstools (v.0.1.0)[54].

### Genomic segmentation into multisample homogenous blocks

We developed and implemented a multichannel dynamic Pprogramming segmentation algorithm to divide the genome into continuous genomic regions (blocks), showing homogeneous methylation levels across multiple CpGs for each sample[54]. A generative probabilistic model is used, each block inducing a Bernoulli distribution with some $\theta_i^k$, where $i$ is the block index and $k$ the sample index ($k = 1,...,K$), and each observation (occurence of one CpG at one sequenced fragment) is represented by a random variable sampled i.i.d. (independent and identically distributed) from the same beta value Ber $\theta_i^k$. The log-likelihood of all sequencing data is the sum of log-likelihoods across all blocks, each decomposing as the sum of log-likelihoods across all samples. The log-likelihood of the $i$th block can therefore be formalized as:

$$\text{score}(\text{block}_i) = ll_i = \Sigma^K_{k=1}((N_C)_i^k \times \log(\hat{\theta}_i^k) + (N_T)_i^k \times \log(1 - \hat{\theta}_i^k))$$

where $(N_C)_i^k$, $(N_T)_i^k$ is the number of methylated and unmethylated observations, respectively, in the $i$th block in the $k$th sample, whereas $\hat{\theta}_i^k$ marks a Bayes estimator of the Bernoulli distribution parameter, calculated with $a_C$, $a_T$ pseudocounts for each block/sample:

$$\hat{\theta}_i^k = \frac{(N_C)_i^k + \alpha_C}{(N_C)_i^k + (N_T)_i^k + \alpha_C + \alpha_T}$$

These hyperparameters are used for regularization, to control the trade-off between overfitting (shorter blocks) and generalization (longer blocks). Dynamic programming is then used to find the optimal segmentation across the genome. Briefly, we maintain a 1 × N table $T$ ($N = 28,217,448$ CpGs) for optimal segmentation scores across all prefixes. Specifically, $T[i]$ holds the score of the optimal segmentation of all CpG sites from 1 through to i, and $T[N]$ holds the final, optimal, score across the entire genome. The table itself is updated sequentially from 1 to $N$, where the optimal segmentation up to position $i$ is achieved by the addition of a new block to a shorter optimal segmentation (for example, up to position $i'$):

$$T[i] = \max_{i' < i}\{T[i'] + \text{score}(\text{block}[i' + 1, \ldots, i])\}$$

For this, all previous optimal segmentations are considered and a new block is added from position ($i' + 1$) to position $i$ (with a maximal block size of 5,000 bp). The combination that maximizes log-likelihood is selected as the optimal segmentation from 1 to $i$, and the start index of the last block is recorded in a traceback table. Once the score of optimal segmentation is calculated in $T[N]$, the traceback table is used to retrieve the full segmentation. An upper bound on block length (5,000 bases) is set to improve running times and each chromosome is run separately. The linear distance between consecutive CpGs is ignored under this model. The model and segmentation algorithm are further described in Supplementary Information.

## Segmentation and clustering analysis

We segmented the genome into 7,104,162 blocks using wgbstools (with parameters 'segment --max_bp 5000') with all of the 205 samples as reference, and retained 2,099,681 blocks covering at least four CpGs. For hierarchical clustering (Fig. 2) we selected the top 1% (20,997) blocks showing the highest variability in average methylation across all samples. Blocks with sufficient coverage of at least ten observations (calculated as sequenced CpG sites) across two-thirds of the samples were further retained. We then computed the average methylation for each block and sample calculated using wgbstools (--beta_to_table -c 10), marked blocks with fewer than ten observations as missing values and imputed their methylation values using sklearn KNNImputer (v.0.24.2)[55]. The 205 samples were clustered with the unsupervised agglomerative clustering algorithm[23], using scipy (v.1.6.3)[56] and L1 norm. The fanning diagram was plotted using ggtree (v.2.2.4)[57].

## Cell-type-specific markers

The 205 atlas samples were divided into 51 groups by cell type, yielding 39 basic groups and 12 composite supergroups (Supplementary Table 3). We then performed a one-versus-all comparison to identify differentially methylated blocks unique for each cell type. For this we used wgbstools' 'find_markers' function to first identify blocks covering at least five CpGs (length 10–1,500 bp) to calculate the average methylation per block/sample and rank the blocks according to the difference in average methylation between target samples versus all other samples. To allow some flexibility, this difference was computed (for unmethylated markers) as the difference between the 75th percentile in target samples (typically allowing one outlier) versus the 2.5th percentile in the background group (typically allowing about five outlier samples). For methylated markers, this was computed as the difference between the 25th and 97.5th percentiles (Supplementary Information). Low-coverage blocks (fewer than 25 observations), in which the estimation error of average methylation was around 10%, were replaced by a default value of 0.5 which is neither unmethylated nor methylated, thus reducing the block's methylation difference and downgrading its rank. For cell type-specific markers we selected the top 25 per cell type, for a total of 1,246 markers (Supplementary Table 4a).

Atlases for 450K/EPIC, RRBS and hybrid capture panels were identified similarly while examining a subset of genomic regions, overlapping various probe sets or genomic regions (-b option). Chromatin analysis was performed on the top 250 markers per cell type (total of 11,713 markers; Supplementary Table 4b). Motif analysis was performed on the top 1,000 markers per cell type (total of 50,286 markers; Supplementary Table 4b) using the difference between the 25th and 75th percentile, to allow putative enhancers unmethylated in additional cell types.

## Enrichment for gene set annotations

Analysis of gene set enrichment was performed using GREAT[31]. For each cell type we selected the top 250 differentially unmethylated regions and ran GREAT via batch web interface using default parameters. Enrichments for 'Ensembl Genes' were ignored, and a significance threshold of binomial false discovery rate ≤0.05 was used.

## Enrichment for chromatin marks

For each cell type we analysed the top 250 differentially unmethylated regions versus published ChIP–seq (H3K27ac and H3K4me1) and DNase sequencing from the Roadmap Epigenomics project (downloaded from ftp.ncbi.nlm.nih.gov/pub/geo/DATA/roadmapepigenomics/by_experiment and http://egg2.wustl.edu/roadmap/data/byDataType/dnase/BED_files_enh) in bigWig and bed formats. These include E032 for B cell markers, E034 for T cell markers, E029 for monocyte/macrophage markers, E066 for liver hepatocytes, E104 for heart cardiomyocytes and fibroblasts and E109 and E110 for gastric/small intestine/colon[4].

Annotations for chromHMM were downloaded (15-states version) from https://egg2.wustl.edu/roadmap/data/byFileType/chromhmmSegmentations/ChmmModels/coreMarks/jointModel/final[3], and genomic regions annotated as enhancers (7_Enh) were extracted and reformatted in bigWig format. Raw single-cell ATAC–seq data were downloaded from GEO GSE165659 (ref. [32]) as 'feature' and 'matrix' files for 70 samples. For each sample, cells of the same type were pooled to output a bedGraph file, which was mapped from hg38 to hg19 using UCSC liftOver[58]. Overlapping regions were dropped using bedtools (v.2.26.0)[59]. Finally, bigWig files were created using bedGraphToBigWig (v.4)[60]. Heatmaps and average plots were prepared using deepTools (v.3.4.1)[61], with the functions 'computeMatrix', 'plotHeatmap' and 'plotProfile'. We used default parameters except for 'referencePoint=center', 15 kb margins and 'binSize=200' for ChIP–seq, DNaseI and chromHMM data, and 75 kb margins with 'binSize=1000' for ATAC–seq data.

## Motif analysis

For each cell type we analysed the top 1,000 differentially unmethylated regions for known motifs (Supplementary Table 6a) using the HOMER function 'findMotifsGenome.pl', with parameters '-bits' and '-size 250'[39]. Similar analyses were performed for the unmethylated regions in each cell type (Supplementary Table 6b), as well as unmethylated regions overlapping H3K27ac, but not H3K4me3, peaks (Supplementary Table 6c).

## Methylation marker–gene associations

For each cell-type-specific marker we identified all neighbouring genes up to 500 kb apart. We then examined the expression levels of these genes across the GTEx dataset covering 50 tissues and cell types[62]. We then standardized the expression of each gene across all conditions, by replacing expression values with standard deviations ($z$-scores) above/below the average expression of that gene across samples. This was followed by column-wise standardization in which the relative enrichment of a gene under a given condition is normalized by the enrichment of other genes under that condition. This highlighted the most overexpressed genes for each tissue. We then classified each 'marker–gene–condition' combination as tier 1: distance ≤5 kb, expression ≥10 TPM and $z$-score ≥1.5; tier 2: same as tier 1 but with distance ≤50 kb; tier 3: up to 750 kb, expression ≥25 TPM and $z$-score ≥5; and tier 4: same as tier 3 but with $z$-score ≥3.5.

## A catalogue of unmethylated loci and putative enhancers for each cell type

For each genomic region (blocks of at least four CpGs), and for any of the 39 cell type groups, fragments with at least four CpGs from all replicates were merged and classified as either U (fragment-level methylation 15% or less), M (at least 85%) or X (over 15% but below 85%). The percentage of U fragments was then calculated using 'wgbstools homog --threshold .15,.85', and blocks with at least 85% unmethylated fragments retained. These blocks were overlapped with genomic features based on UCSC hg19 annotations, including CpG islands and transcriptional start site regions (up to 1 kb from a gene start site). We also used narrowPeak annotations downloaded from Roadmap[4] and ENCODE project[5] (accessions listed in Supplementary Table 6d). hg38 bed files were converted to hg19 using liftOver[58]. For putative enhancers, nonpromoter active regulatory regions were defined as those overlapping H3K27ac, but not H3K4me3, peaks under matching conditions. TF binding sites were downloaded from JASPAR 2022 (ref. [63]).

## Interindividual variation in cell type methylation

We define a similarity score between two samples as the fraction of blocks containing at least three CpGs and at least ten binary observations (sequenced CpG sites) in which the average methylation of the two samples differs by at least 0.5. Only cell types with $n ≥ 3$ FACS-sorted replicates from different donors are considered (136 samples in total).

## CTCF ChIP–seq analysis

CTCF ChIP–seq data were downloaded from the ENCODE project[5] as 168 bigWig files, covering 61 tissues/cell types (hg19). Samples of the same cell type were averaged using multiBigwigSummary (v.3.4.1)[61].

## Endodermal marker analysis

All 892 endodermal hypomethylated markers were found using wgbstools function 'find_markers' (v.0.2.0), with parameters '--delta_quants 0.4 --tg_quant 0.1 --bg_quant 0.1' (ref. [54]). For endoderm-derived epithelium, 51 samples were compared with 103 nonepithelial samples from mesoderm or ectoderm. Blocks were selected as markers if the average methylation of the 90th percentile of the epithelial samples was lower than the tenth percentile of the nonepithelial samples by at least 0.4.

## UXM fragment-level deconvolution algorithm

We developed a fragment-level deconvolution algorithm: each fragment was annotated as U (mostly unmethylated), M (mostly methylated) or X (mixed) depending on the number of methylated and unmethylated CpGs[64]. We then calculated, for each genomic region (marker) and across all cell types, the proportion of U/X/M fragments with at least $k$ CpGs. Here we used $k = 4$ and thresholds of less than or equal to 25% methylated CpGs for U reads, and more than or equal to 75% methylated CpGs for M reads. We then constructed reference atlas $A$ with 1,232 regions (top 25 markers per cell type), in which the $A_{i,j}$ cell holds the U proportion of the $i$th marker in the $j$th cell type. Given an input sample, the U proportion at each marker is computed to form a $1,232 \times 1$ vector $b$. Then, NNLS is applied to infer coefficient vector $x$ by minimizing $|A \times x - b|_2$ subject to non-negative $x$, normalized to $\sum_j x_j = 1$. Alternatively, each marker can be weighed differently based on fragment coverage in the input sample. For this, $b$ can be defined as the number of U fragments in each region and the rows of $A$ similarly multiplied by $C_i$, the total number of fragments in each region, thus minimizing $|diag(C) \times A \times x - b|_2$. Additional details are available in Supplementary Information.

## In silico simulation of WGBS deconvolution

Simulated mixtures were performed for cardiomyocytes ($n = 4$), bladder epithelium ($n = 5$), breast epithelium ($n = 7$), endothelial cells ($n = 19$) and erythrocyte progenitors ($n = 3$) in a leave-one-out manner. For this, one sample was held out and segmentation and marker selection (25 per cell type) were rerun using the remaining 204 samples. We then simulated mixtures by sampling and mixing reads from the held-out sample at 10, 3, 1, 0.3, 0.1, 0.03 and 0% into a background of leukocyte samples. This was repeated ten times. Finally, mixed samples were analysed using the UXM fragment-level algorithm with markers from the reduced (204) atlas, using fragments with at least three CpGs. Merging, splitting and mixing of reads were performed using wgbstools (v.0.1.0)[54].

Array-based analysis was performed by computing, for each mixed set of fragments, average methylation levels across each of around 480,000 CpG sites present in the 450K array ('wgbstools beta_to_450k'). We then deconvolved these data according to the method of Moss et al.[28] (https://github.com/nloyfer/meth_atlas).

We also simulated four-way mixtures in which background plasma methylomes were simulated as a combination of 90% fragments from leukocytes, 7.5% from a vascular endothelial sample and 2.5% from a hepatocyte sample. As described above, this was done by holding out the three samples (for example, cardiomyocytes, endothelial cells and hepatocytes) and then rerunning segmentation and marker selection on the ($202 = 205 - 3$) remaining samples, to obtain a set of markers that was then used for fragment-level deconvolution of mixtures.

## WGBS deconvolution

Leukocytes and matching plasma samples ($n = 23$) were processed as described above and analysed using the WGBS methylation atlas, including 1,246 markers plus (for plasma samples) an additional 25 megakaryocyte markers. Fifty-two plasma samples from 28 patients with SARS-CoV-2 (ref. [44]) downloaded as FASTQ files were processed as described above. Because of the low coverage (1–2×) of these samples, we extended the marker set from the top 25 to the top 250 markers per cell type (Supplementary Table 4b), and also included 250 megakaryocyte markers[65]. Roadmap[4] and ENCODE[5] samples were processed as described above and analysed using the UXM algorithm.

## Deconvolution of 450K array data

Previously published 450K array data were downloaded from either The Cancer Genome Atlas (lung and breast biopsies)[49,50] or GEO accession no. GSE62640 (ref. [48]) and deconvoluted with meth_atlas NNLS software (https://github.com/nloyfer/meth_atlas) using our array-adapted atlas (Supplementary Table 12). Breast biopsies were grouped using PAM50 classifications[66].

## Reporting summary

Further information on research design is available in the Nature Portfolio Reporting Summary linked to this article.

## Data availability

DNA methylation data are available in formats bigWig (position and average methylation across 28,217,448 CpGs) and beta (a similar wgbstools-compatible binary format) at GEO (accession no. GSE186458). BigWig and beta files for hg38 are also available. Fragment-level information (in pat format, including CpG starting index, methylation pattern of all covered CpGs and number of fragments with exact multiCpG pattern) are also available. Raw fastq files have been deposited at the European Genome-phenome Archive (EGA) under study accession number: EGAS00001006791 and can be downloaded upon request to EGA (through the atlas Data Access Committee).

## Code availability

Code is available at github.com/nloyfer/wgbs_tools and github.com/nloyfer/UXM_deconv.

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

**Acknowledgements** We thank H. Cedar and N. Friedman for insightful discussions. We also thank members of the Dor, Kaplan and Rosenfeld laboratories. This work was supported by grants from GRAIL, Alzheimer's Drug Discovery Foundation, Human Islet Research Network (nos. HIRN UC4DK116274 and UC4DK104216), the Ernest and Bonnie Beutler Research Program of Excellence in Genomic Medicine, The Alex U Soyka pancreatic cancer fund, The Israel Science Foundation, the Waldholtz/Pakula family, the Robert M. and Marilyn Sternberg Family Charitable Foundation, the Helmsley Charitable Trust and DON Foundation (to Y.D.), Israel Science Foundation (no. 1250/18 to T.K.) and the Center for Interdisciplinary Data Science Research (to T.K., Y.D. and B.G.). N.L. was supported by CIDR Data Science and Leibniz fellowships. Y.D. holds the Walter and Greta Stiel Chair and Research Grant in Heart Studies.

**Author contributions** A.A., G.C., R.S., B.G., T.K. and Y.D. conceived and initiated the project and designed the experiments. D.B., D.S., S.P., U.D., G.R., O.O., N.H., E.C., A. Pikarsky, A. Khalaileh, G.Z., R.G., M.A.G., I.M., N.S., A. Korach, O.W., U.I., E.E., V.Y., Y.S., D.R.G., K.L.S., H.D., P.A., A.M.J.S. and M.G. provided materials. J. Magenheim, A. Peretz, A. Klochendler, I.F.-F., M.H., T.P. and Z.D. performed experiments. G.C., J.B., H.A., P.M., S.N., O.V. and A.J. conducted sequencing and analysis. N.L., S.S.-P., J. Moss and T.K. developed and performed computational analyses. N.L., Y.D. and T.K. wrote the manuscript.

**Competing interests** This work was supported by GRAIL, Inc. G.C., J.B., A.A., O.V. and A.J. are employees, shareholders and/or founders at GRAIL, Inc. J.M., J.M., I.F.-F., R.S., Y.D., B.G. and T.K. have filed patents on cfDNA analysis technology. The remaining authors declare no competing interests.

**Additional information**
**Correspondence and requests for materials** should be addressed to Yuval Dor, Benjamin Glaser or Tommy Kaplan.

# A    Cell type-specific human methylation atlas

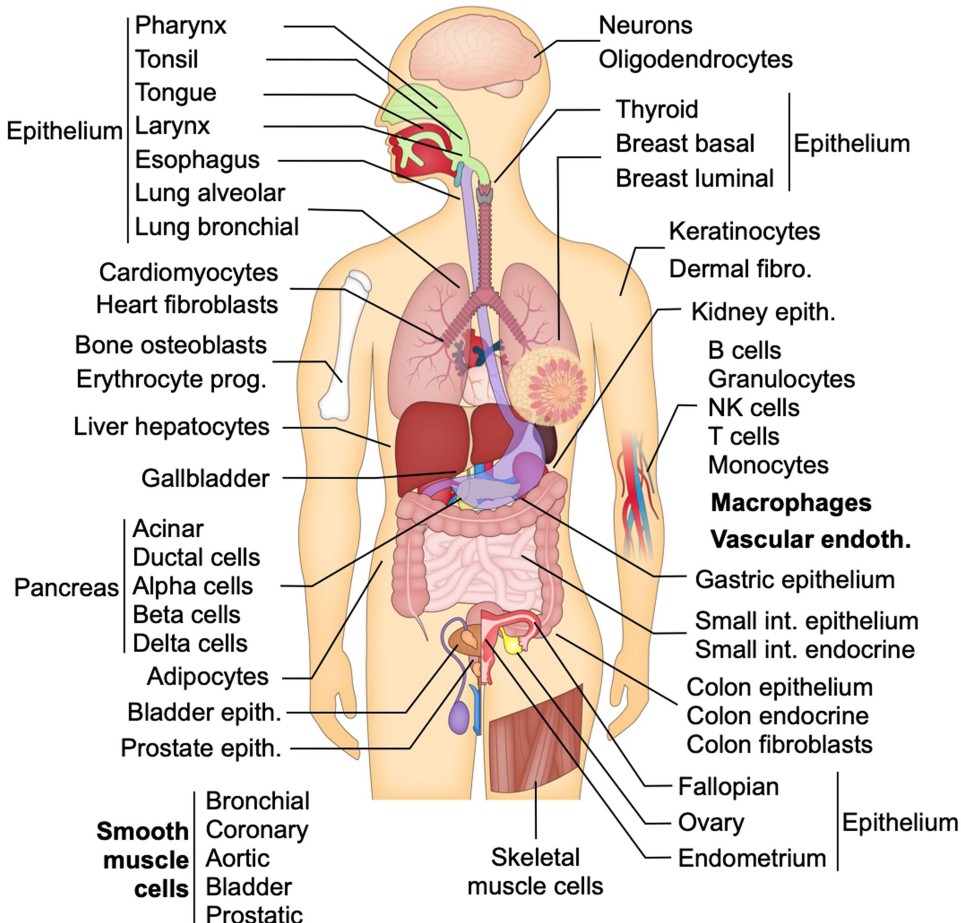

**Extended Data Fig. 1 | A human whole-genome DNA methylation atlas of healthy cell types.** 205 healthy samples were obtained from adult humans, isolated and deeply sequenced (WGBS, mean depth ≥30x), to form a comprehensive human cell-type-specific methylation atlas.

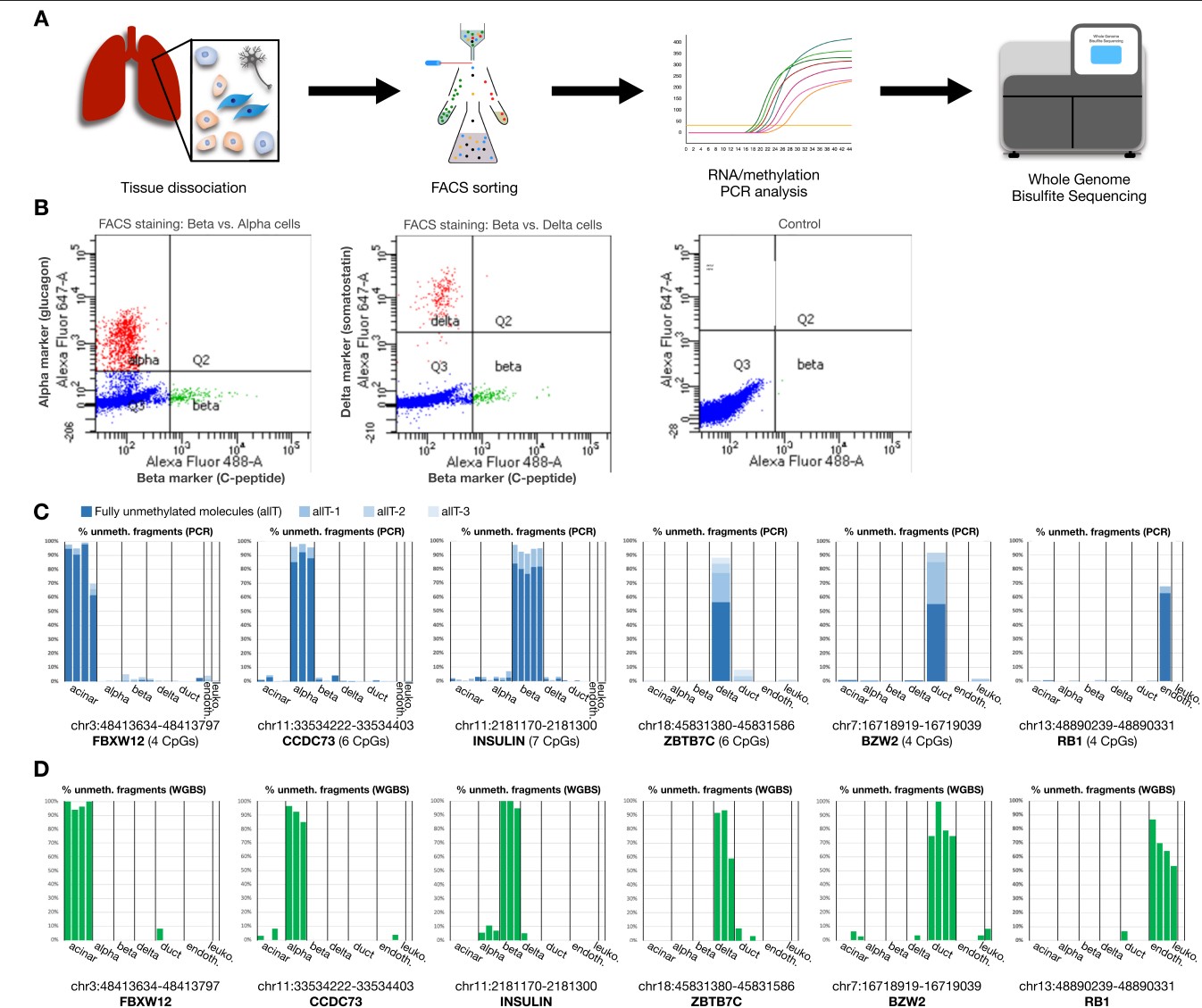

**Extended Data Fig. 2 | Sample preparation and purity. (A)** Fresh tissue was obtained at surgery and dissociated (optimized per tissue type), then incubated with antibodies, and FACS-sorted. Sorted cells were analysed using qRT-PCR for key cell-type-specific genes, or targeted PCR for cell-type-specific DNA methylation markers. DNA methylation was also analysed using whole-genome bisulfite sequencing. **(B)** Example of FACS sorting for pancreatic endocrine cell types. Left panel: staining for the beta cell marker C-peptide (x-axis) versus alpha cell marker glucagon (y-axis). Note that no double positive cells are observed. Centre panel: staining for c-peptide (x-axis) versus delta cell marker somatostatin (y-axis). Right panel: unstained control (only fluorescent secondary antibodies added, no primary antibodies). **(C)** Fragment-level validation of sample purity using targeted PCR. Cell-type-specific markers

were designed using pre-existing 450K data, covering 4–7 several neighbouring CpGs. Shown is the percentage of unmethylated molecules in each cell type (including endothelial cells and leukocytes). Colour gradient fades from fully unmethylated molecules (allT), through those unmethylated in all but one CpG (allT-1), etc. Amplicon locations are reported in hg19, for acinar cells, alpha, beta, delta, duct, and endothelial markers (from left to right). **(D)** Fragment-level validation of the same locations, using the atlas WGBS data. Y-axis marks the percentage of unmethylated fragments (with ≥4 CpGs). As these markers show, approximately 90% of molecules in that target cell type are unmethylated, compared with less than 5% in other cell types, thus emphasizing the purity of the DNA methylation atlas using a set of independently selected DMRs.

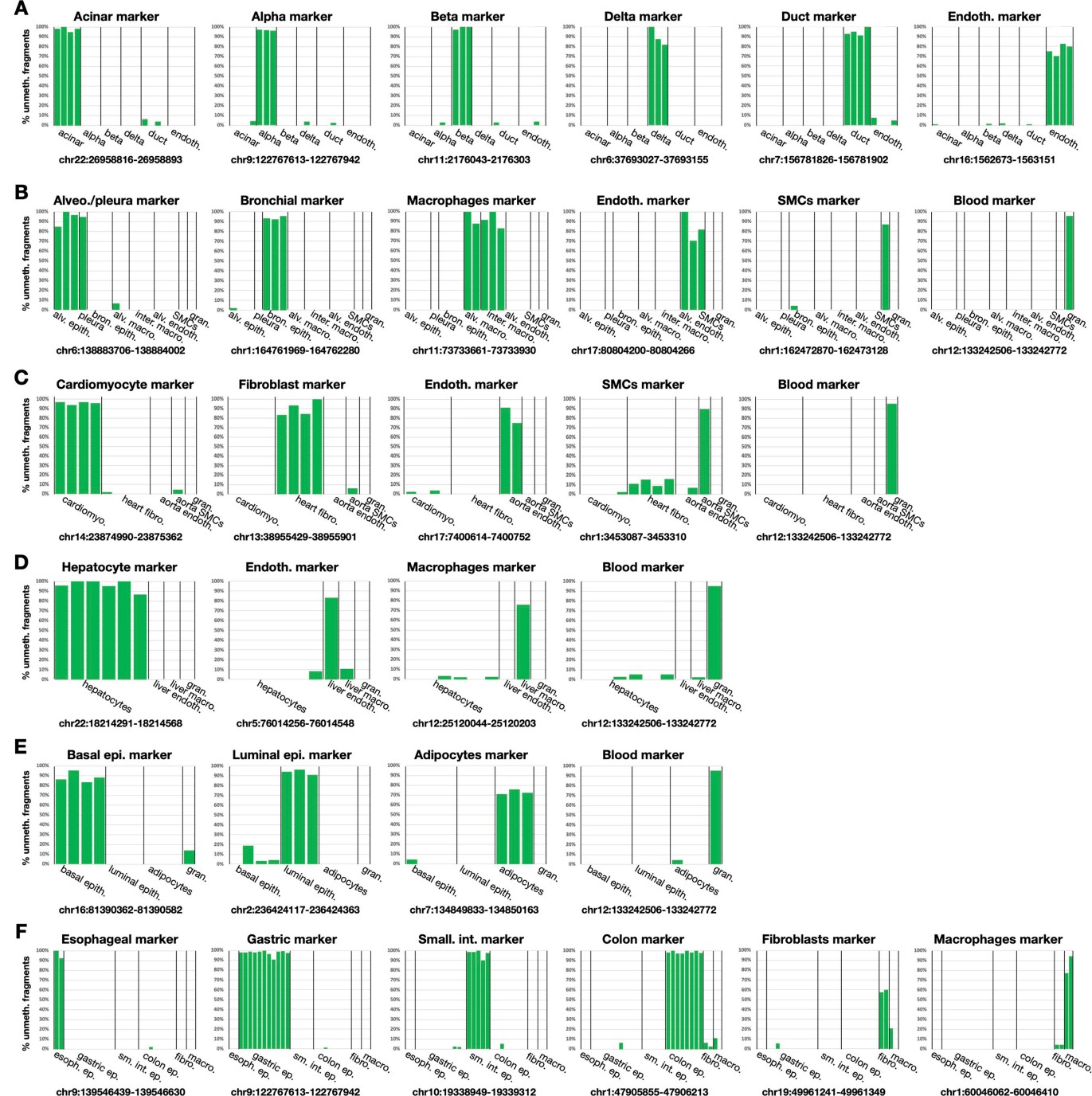

**Extended Data Fig. 3 | Purity estimation for pancreas, lung, heart, liver, breast, and GI using atlas markers.** The percent of unmethylated fragments (y-axis) among fragments of ≥4 CpGs from selected differentially methylation markers could serve as an (under-) estimate of the atlas purity. Here we show one such marker for each cell type, selected from the top 25 markers, and use fragment-level analysis to demonstrate the purity in the target cell type compared to other cell types from the same tissue or environment. (**A**) Pancreas. (**B**) Lung. (**C**) Heart. (**D**) Liver. (**E**) Breast. (**F**) GI tract. For most cell types, 90% of the molecules in the target cell types are unmethylated, compared with less than 5% of other types. This is an under-estimation, as some heterogeneity could occur in each cell type, reflecting stochastic noise, cellular states, age, or environmental changes.

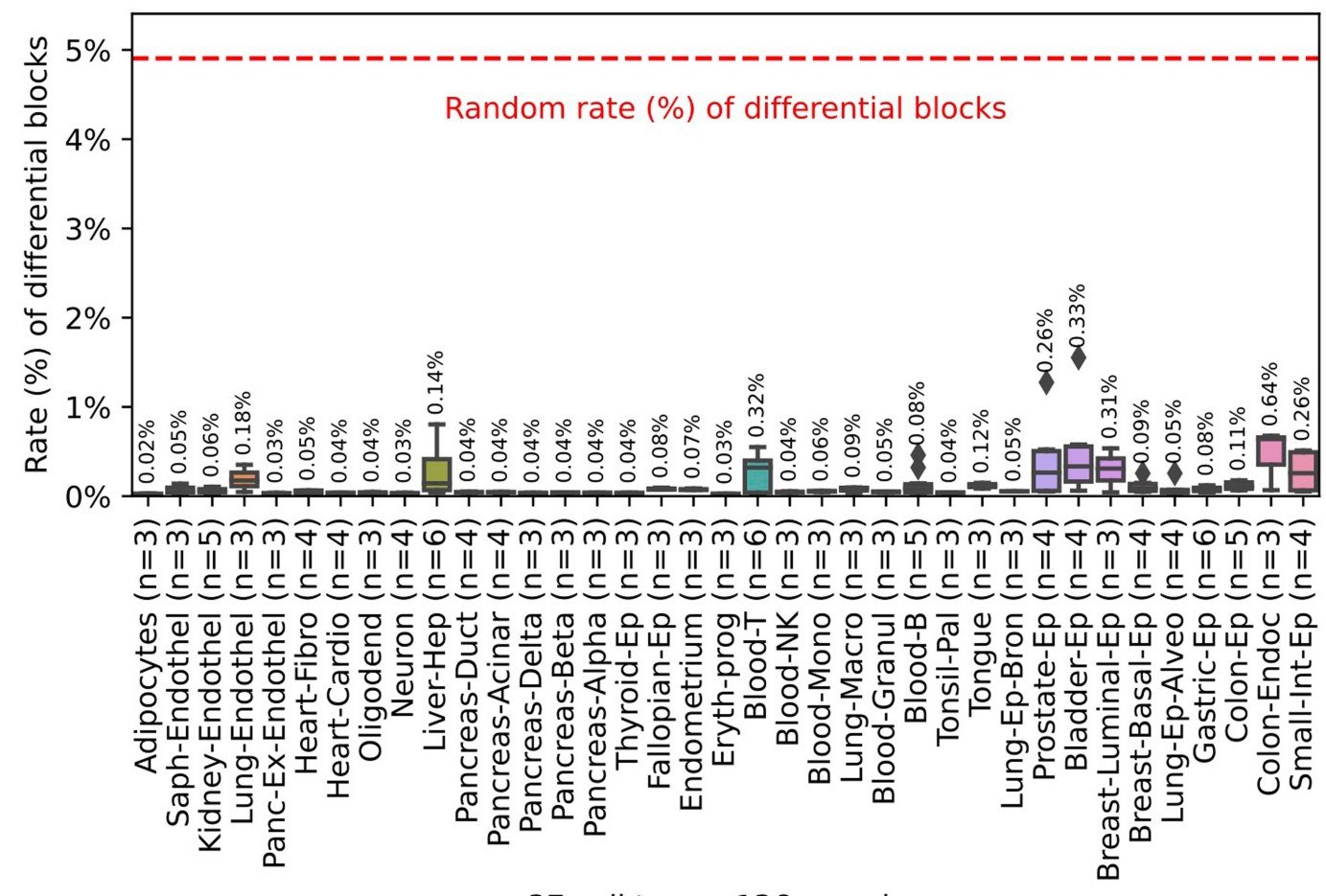

**Extended Data Fig. 4 | Biological replicates of the same cell type, from different individuals show a surprisingly low rate of differentially methylated blocks.** We focused on 37 cellular subtypes with n≥3 replicates (e.g. endothelial cells from a specific tissue) and measured the average percentage of methylation blocks (≥3 CpGs) that differ in their methylation by 50% (absolute delta beta), across replicates (shown as Y-axis). Nearly all cellular subtypes (36/37) differ by ≤0.5% of blocks suggesting a very high degree of conservation among replicates. Dotted red line marks the average number of differential blocks between two random samples of different cell types (4.9%). Box plots mark median and interquartile range (IQR), with 1.5*IQR whiskers.

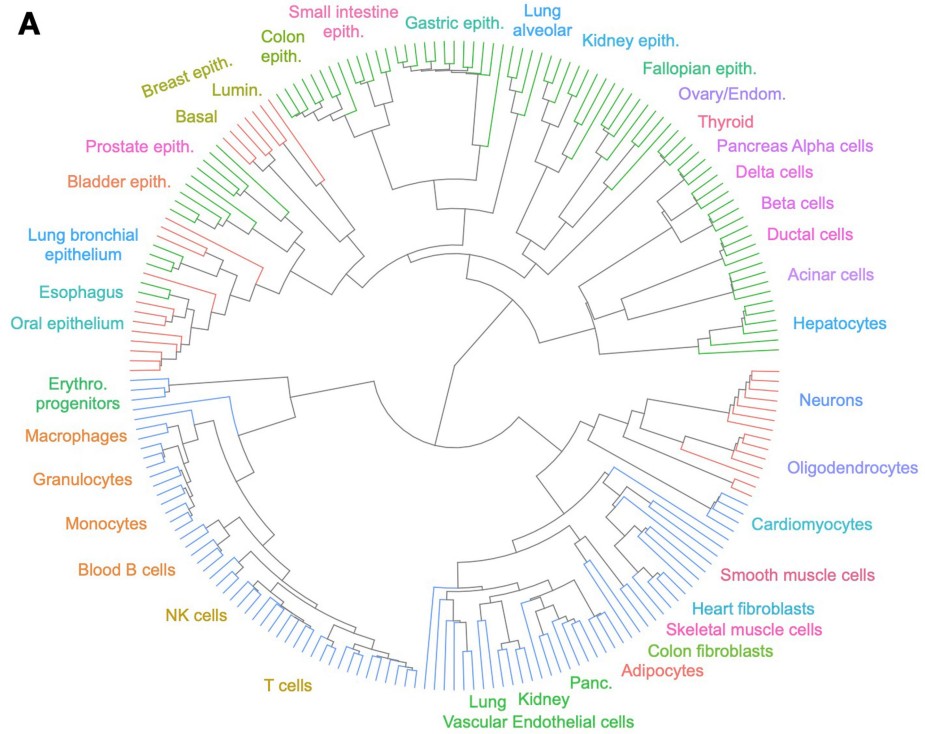

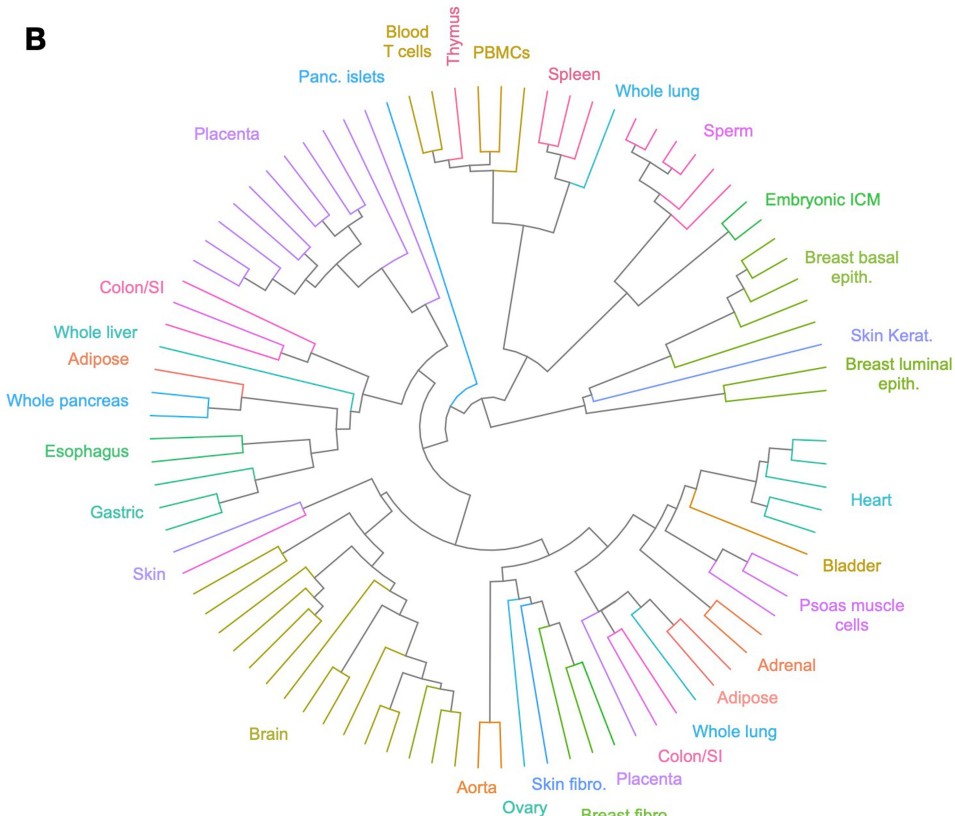

**Extended Data Fig. 5 | Unsupervised agglomerative clustering of human methylomes. (A)** Same as Fig. 2, coloured by developmental lineage from germ layers, including endoderm (green), mesoderm (blue), and ectoderm (red). **(B)** same as Fig. 2, for Roadmap Epigenomics DNA methylation atlas.

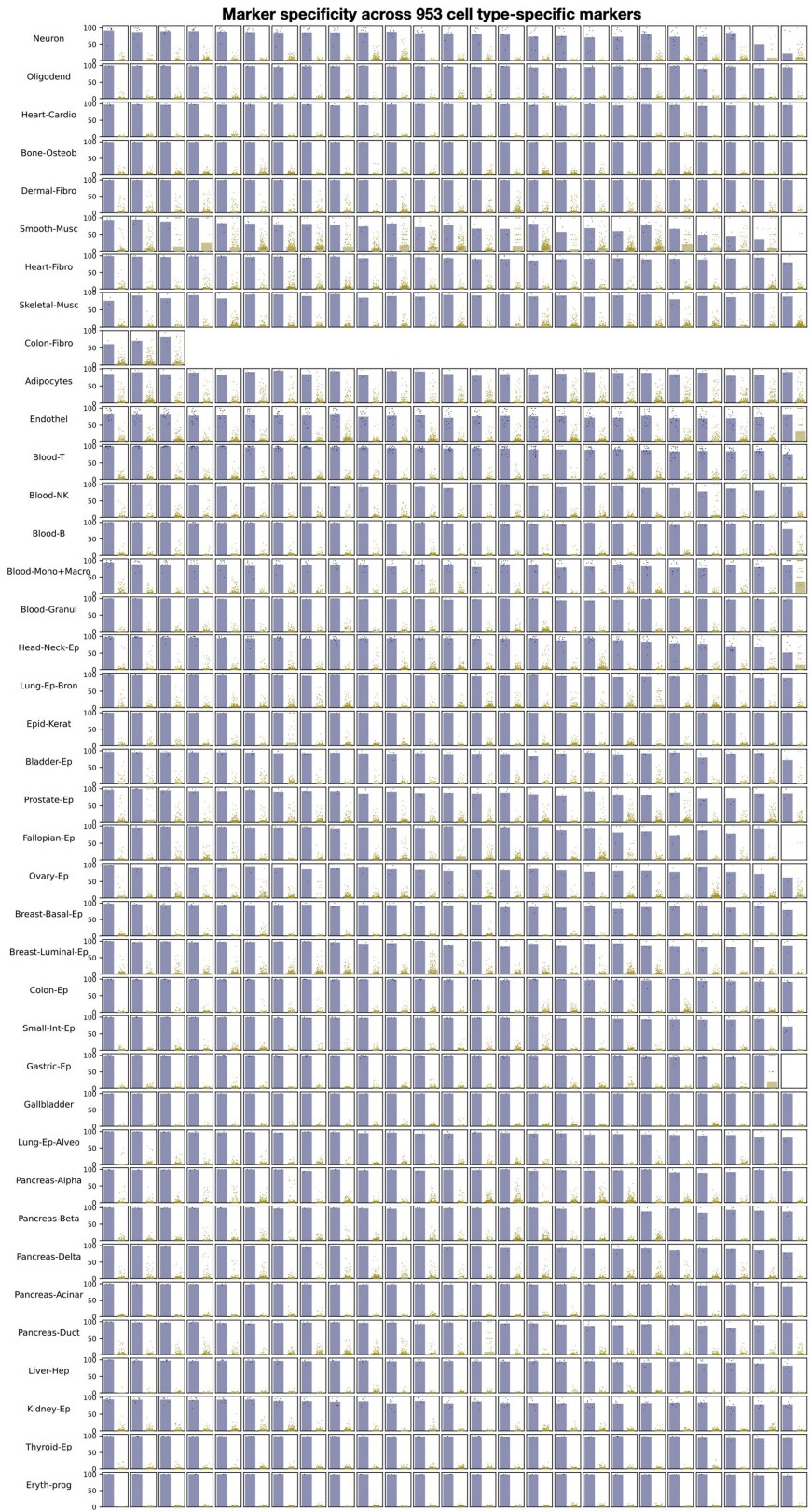

**Extended Data Fig. 6 | Marker specificity across 953 cell-type-specific markers.** For every cell type (row), we plot each of the top 25 markers (shown as boxes). For each marker, we compare the percentage of unmethylated fragments (≥3 CpGs) in the target samples (blue dots) versus their percentage in background samples (golden dots). Blue and golden bars plot the average proportion across all target and background samples, respectively.

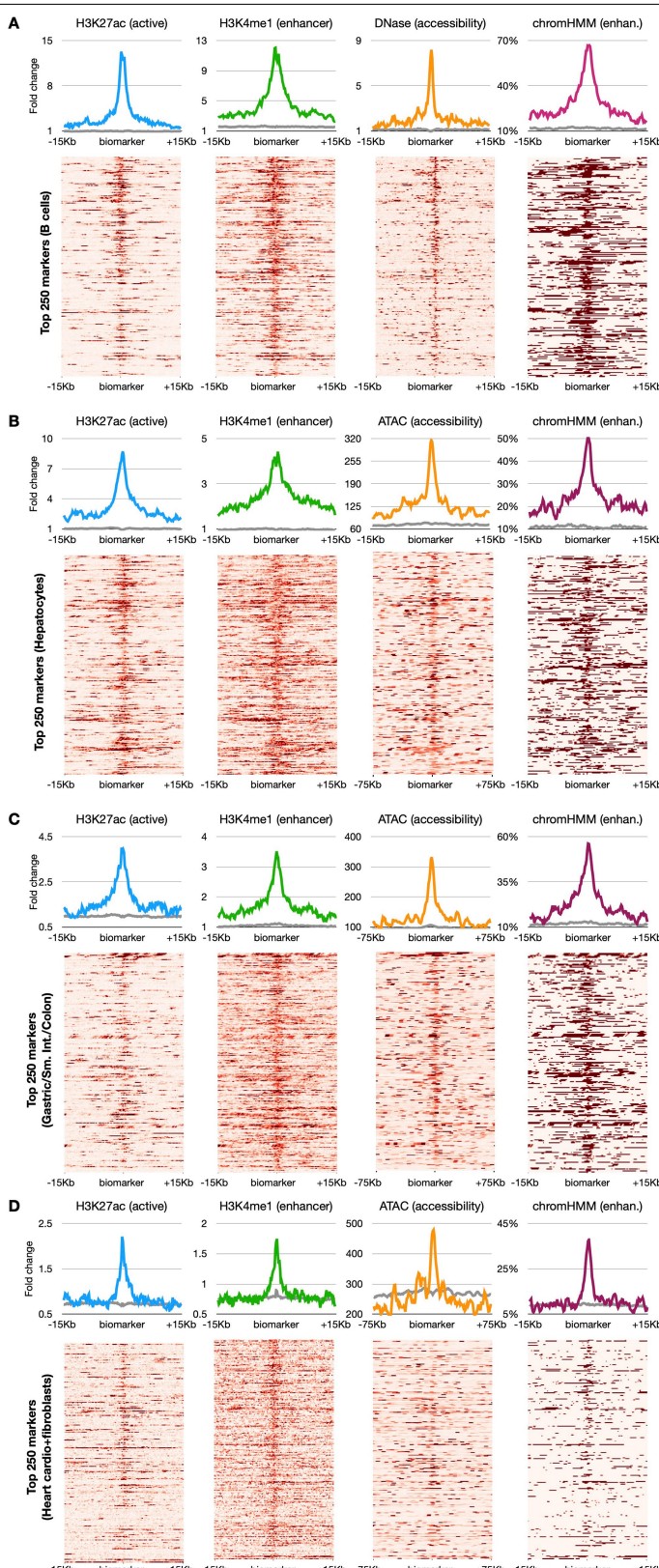

**Extended Data Fig. 7 | Markers of putative enhancers in other atlas cell types.** Including top 250 unmethylated markers for B cells (top left), hepatocytes (top right), gastric/small intestine/colon epithelium (bottom left), and cardiomyocytes/heart fibroblasts (bottom right). Grey lines mark the same ChIP-seq/ATAC/DNase/chromHMM signal, averaged across all 11,371 unmethylated markers (top 250 per cell type).

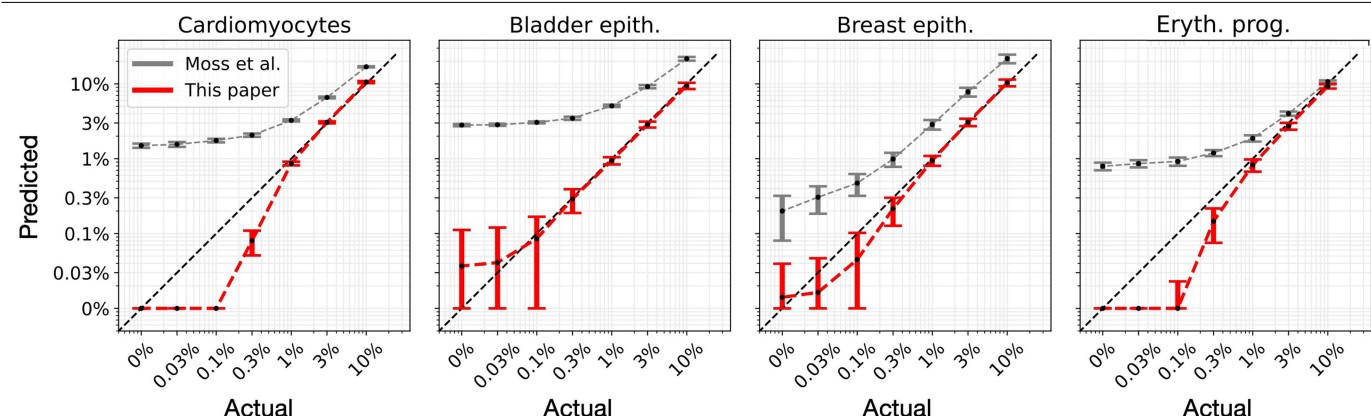

**Extended Data Fig. 8 | Fragment-level analysis of *in silico* mixes.** Shown are *in silico* simulations for four cell types, which are computationally mixed at various proportions with a plasma-like mixture of 90% leukocytes, 7.5% vascular endothelial cells, and 2.5% hepatocytes. Each mixture was analysed using our atlas (red), and compared to Moss et al. (grey). Box plots show average contribution in 10 simulations, with 1 SD error bars.

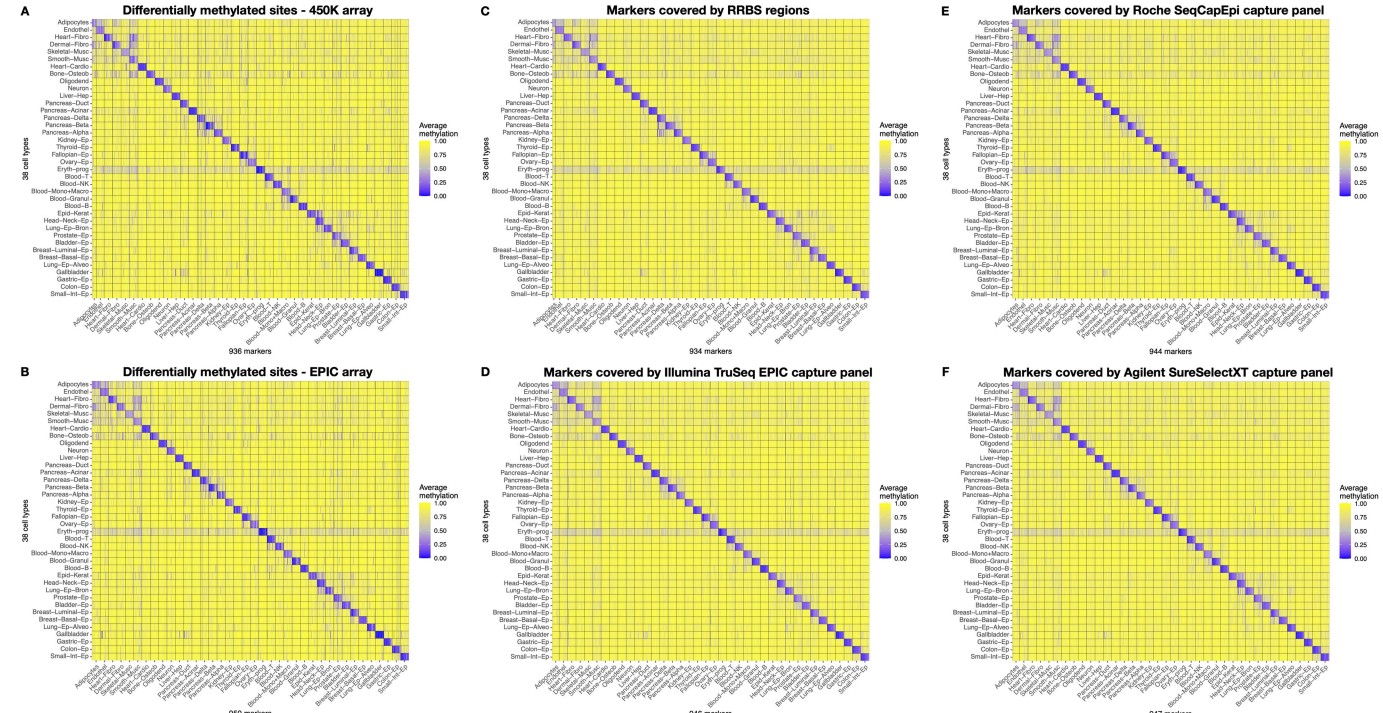

**Extended Data Fig. 9 | Specialized DNA methylation atlases for the analysis of (A) Illumina BeadChip 450K array, (B) EPIC array, (C) RRBS sequencing data, (D) Illumina TruSeq EPIC capture panel, (E) Roche SeqCapEpi capture panel, and (F) Agilent SureSelectXT capture panel .**

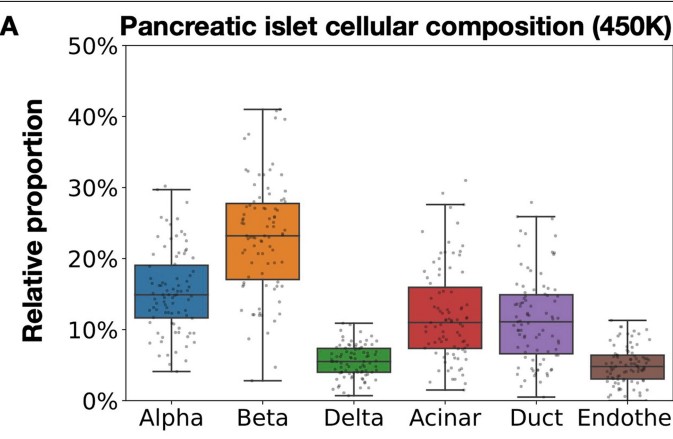

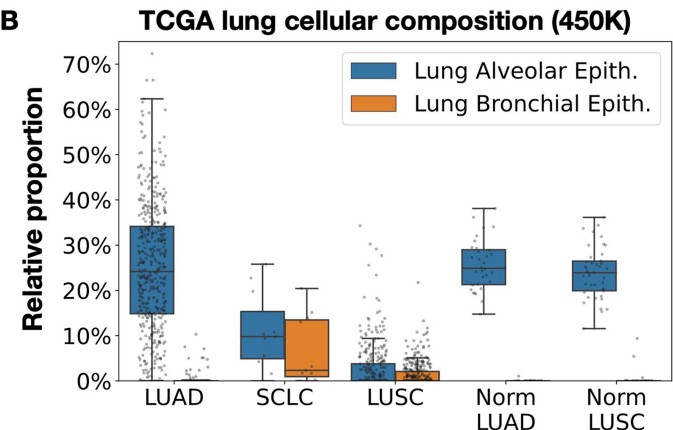

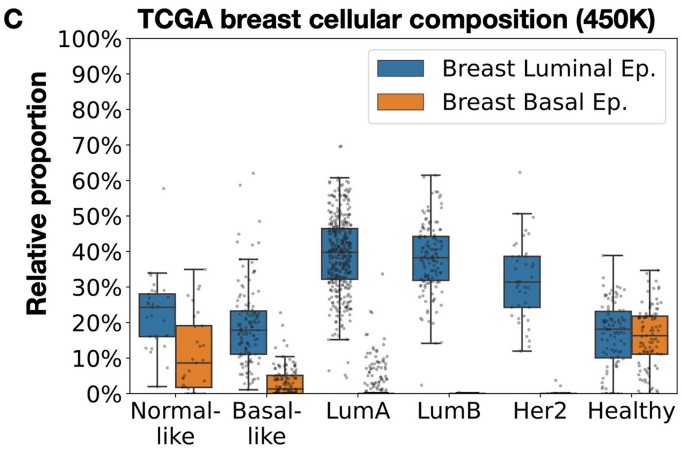

**Extended Data Fig. 10 | Deconvolution of previously published 450K DNA methylation array data.** (**A**) Deconvolution of pancreatic islet methylomes. Methylation arrays from 53 male and 34 female non-diabetic donors[48] were analysed atlas methylomes, revealing detailed cellular composition including previously uncharacterized alpha and delta cells. No statistically significant sex differences in cellular composition were observed. (**B**) Analysis of 865 pulmonary methylomes from TCGA[49]. WGBS-based markers for lung alveolar epithelium and lung bronchial epithelium cells reveal differential cell populations in 443 LUAD, 11 SCLC, 337 LUSC, 32 normal adjacent (LUAD), and 42 normal adjacent (LUSC) lung methylomes. Note that only alveolar cell DNA is identified in lung adenocarcinomas, while small cell lung cancer and squamous cell carcinomas contain also bronchial DNA, consistent with the presumed cellular origins of each type of lung cancer. Note that epithelial cells are a minority in both lung adenocarcinoma and normal lungs. This is probably due to the abundance of stromal cells in bulk preparations of either normal lungs or lung cancers. (**C**) DNA methylation from 721 cancerous and 97 normal breast biopsies from TCGA. WGBS-based markers for breast luminal and basal epithelial cells were used to study the cellular composition in TCGA[50], which were classified into five subtypes using PAM50, a 50-gene expression-based classification[66]. Different cell composition is observed for normal-like, basal-like, luminal A, luminal B, and Her2-enriched PAM50 subtypes, compared to healthy breast biopsies. The low fraction of breast basal cells in breast cancer is likely to result from the abundance of non-epithelial cells in both the normal breast and breast cancer. Box plots mark median and interquartile range (IQR), with 1.5*IQR whiskers.

# Reporting Summary

## Statistics

For all statistical analyses, confirm that the following items are present in the figure legend, table legend, main text, or Methods section.

| n/a | Confirmed | |
|---|---|---|
| ☐ | ☒ | The exact sample size (*n*) for each experimental group/condition, given as a discrete number and unit of measurement |
| ☐ | ☒ | A statement on whether measurements were taken from distinct samples or whether the same sample was measured repeatedly |
| ☐ | ☒ | The statistical test(s) used AND whether they are one- or two-sided<br>*Only common tests should be described solely by name; describe more complex techniques in the Methods section.* |
| ☐ | ☒ | A description of all covariates tested |
| ☐ | ☒ | A description of any assumptions or corrections, such as tests of normality and adjustment for multiple comparisons |
| ☐ | ☒ | A full description of the statistical parameters including central tendency (e.g. means) or other basic estimates (e.g. regression coefficient) AND variation (e.g. standard deviation) or associated estimates of uncertainty (e.g. confidence intervals) |
| ☐ | ☒ | For null hypothesis testing, the test statistic (e.g. $F$, $t$, $r$) with confidence intervals, effect sizes, degrees of freedom and $P$ value noted<br>*Give P values as exact values whenever suitable.* |
| ☐ | ☒ | For Bayesian analysis, information on the choice of priors and Markov chain Monte Carlo settings |
| ☐ | ☒ | For hierarchical and complex designs, identification of the appropriate level for tests and full reporting of outcomes |
| ☐ | ☒ | Estimates of effect sizes (e.g. Cohen's *d*, Pearson's *r*), indicating how they were calculated |

*Our web collection on statistics for biologists contains articles on many of the points above.*

## Software and code

Policy information about availability of computer code

| Data collection | no software was used |
|---|---|
| Data analysis | bedtools (v 2.26.0)<br>bedGraphToBigWig (V 4)<br>deepTools (V 3.4.1)<br>multiBigwigSummary(V 3.4.1)<br>bwa-meth (V 0.2.0)<br>SAMtools (V 1.9)<br>Sambamba (V 0.6.5)<br>wgbstools (V 0.1.0)<br>sklearn KNNImputer (V 0.24.2)<br>scipy (V 1.6.3)<br>ggtree (V 2.2.4)<br>GREAT (V 4.0.4, https://great.stanford.edu/)<br>HOMER findMotifsGenome.pl (V2)<br>wgbstools (v 0.1.0, https://github.com/nloyfer/wgbs_tools)<br><br>Code is available at github.com/nloyfer/wgbs_tools and github.com/nloyfer/UXM_deconv |

For manuscripts utilizing custom algorithms or software that are central to the research but not yet described in published literature, software must be made available to editors and reviewers. We strongly encourage code deposition in a community repository (e.g. GitHub). See the Nature Portfolio guidelines for submitting code & software for further information.

## Data

Policy information about availability of data

All manuscripts must include a data availability statement. This statement should provide the following information, where applicable:

- Accession codes, unique identifiers, or web links for publicly available datasets
- A description of any restrictions on data availability
- For clinical datasets or third party data, please ensure that the statement adheres to our policy

DNA methylation data is available in bigwig format (position and average methylation across 27,927,160 CpGs), and beta format (a similar wgb tools-compatible binary format) at the GEO, accession GSE186458. Bigwig and beta files for hg38 are also available. Fragment-level information (in pat format, including CpG starting index, methylation pattern of all covered CpGs, and number of fragments with this exact multi-CpG pattern) are also available. Raw fastq can be downloaded upon request to EGA (through the atlas Data Access Committee).

## Human research participants

Policy information about studies involving human research participants and Sex and Gender in Research.

| | |
|---|---|
| Reporting on sex and gender | No gender-related data was collected. Sex data is detailed in Extended Table S1 |
| Population characteristics | We used >200 tissue specimens from consented patients admitted to surgery at Hadassah Medical Center. Detailed information on donors is provided in Extended Table S1. |
| Recruitment | This is not a population-based study. We are defining tissue-specific methylation patterns that are universally conserved among all individuals. Prospective donors were approached, received an explanation and signed informed consent. |
| Ethics oversight | Study was approved by the Helsinki committee of the Hadassah Medical Center. Some cells and tissues were obtained through collaborative arrangements (Extended Table S1). These include pancreatic exocrine and liver samples (cadaveric organ donors, n=5) from Prof. Markus Grompe, Oregon Health & Science University. Adipocytes (subcutaneous adipocytes at time of cosmetic surgery following weight loss; n=3), oligodendrocytes and neurons (brain autopsies, n=14) from Profs. Kirsty L. Spalding and Henrik Druid, Karolinska Institute, Stockholm, and research grade cadaveric pancreatic islets from Prof. James Shapiro, University of Alberta (n=16). In all cases tissues were obtained and transferred in compliance with local laws and after the approval of the local ethics committee on human experimentation. Sixteen cell types were obtained from commercial sources, including 15 from Lonza Walkersville, Walkersville, MD, U.S.A. and one from Sigma Aldrich. Three pancreatic islet preparations were obtained from the Integrated Islet Distribution Program (IIDP, https://iidp.coh.org). |

Note that full information on the approval of the study protocol must also be provided in the manuscript.

# Field-specific reporting

Please select the one below that is the best fit for your research. If you are not sure, read the appropriate sections before making your selection.

☒ Life sciences ☐ Behavioural & social sciences ☐ Ecological, evolutionary & environmental sciences

For a reference copy of the document with all sections, see nature.com/documents/nr-reporting-summary-flat.pdf

# Life sciences study design

All studies must disclose on these points even when the disclosure is negative.

| | |
|---|---|
| Sample size | Based on sample availibility |
| Data exclusions | Healthy samples, Low sequencing coverage |
| Replication | Yes, across individuals. We define a similarity score between two samples as the fraction of blocks containing ≥3 CpGs, and ≥10 binary observations (sequenced CpG sites), where the average methylation of the two samples differs by ≥0.5. Only cell types with n≥3 FACS-sorted replicates from different donors are considered (136 samples in total). |
| Randomization | N/A |
| Blinding | Not relevant |

# Reporting for specific materials, systems and methods

We require information from authors about some types of materials, experimental systems and methods used in many studies. Here, indicate whether each material, system or method listed is relevant to your study. If you are not sure if a list item applies to your research, read the appropriate section before selecting a response.

## Materials & experimental systems

| n/a | Involved in the study |
|---|---|
| ☐ | ☒ Antibodies |
| ☒ | ☐ Eukaryotic cell lines |
| ☒ | ☐ Palaeontology and archaeology |
| ☒ | ☐ Animals and other organisms |
| ☒ | ☐ Clinical data |
| ☒ | ☐ Dual use research of concern |

## Methods

| n/a | Involved in the study |
|---|---|
| ☒ | ☐ ChIP-seq |
| ☐ | ☒ Flow cytometry |
| ☒ | ☐ MRI-based neuroimaging |

# Antibodies

| | |
|---|---|
| Antibodies used | Attached table S1 |
| Validation | Described in attached file "Supplementary Information" |

# Flow Cytometry

## Plots

Confirm that:

☒ The axis labels state the marker and fluorochrome used (e.g. CD4-FITC).

☒ The axis scales are clearly visible. Include numbers along axes only for bottom left plot of group (a 'group' is an analysis of identical markers).

☒ All plots are contour plots with outliers or pseudocolor plots.

☒ A numerical value for number of cells or percentage (with statistics) is provided.

## Methodology

| | |
|---|---|
| Sample preparation | Described in file "Supplementary Information" |
| Instrument | FACS BD Aria |
| Software | BD FACSDiva 8.0.1 |
| Cell population abundance | Described in file "Supplementary Information" |
| Gating strategy | Described in file "Supplementary Information" |

☒ Tick this box to confirm that a figure exemplifying the gating strategy is provided in the Supplementary Information.

