## [Peer Review File · Nature]

Manuscript Title: A DNA methylation atlas of normal human cell types

Reviewer Comments & Author Rebuttals

Reviewer Reports on the Initial Version:

Referees' comments:

Referee #1 (Remarks to the Author):

Loyfer et al report WGBS data for 207 samples capturing 39 cell types. The average coverage is 30x with the lowest being 5x. The authors then identify methylation blocks (2.8M). Samples of the same type are highly correlated. Unsupervised clustering recreated major lineage trees. Differential methylation analysis finds cell type specific unmethylated blocks enriched for TF binding sites and enhancers and reports a small percentage of regions that gain methylation as expected at canonically unmethylated sites (CTCF sites and CGIs). The unique signatures can be used to deconvolute cell mixtures. This is clearly a massive endeavour and major data generation effort.

Such an extensive set of WGBS data is clearly a valuable resource. The sequencing coverage (average 30x) is standard for small sample numbers but likely limiting for large scale comparisons.

Unfortunately, some key numbers are not reported: i) the bisulfite conversion rate determined through CC conversion rates (not spike ins) needs to be provided, ii) the number of CpGs covered at 1x and 10x in each sample.

To the above point, how many CpGs are covered at 10x across all 207 samples?

The current Figures are not very informative, in particular Figures 1-4. Most of these overview panels could be in the supplement and the main text should have better examples that showcase the data and new biology.

The text is often redundant and the discussion is largely a repeat summary of the results.

Overall, the data sets have a lot of potential but currently a very limited number of insights are presented and the few selected highlights are not entirely new and not properly discussed in the context of prior literature. For instance, it is well established that cell- and tissue-specific methylation differences are found at enhancers. Nothing of relevance is reported for the hypermethylated regions. It is expected that unmethylated CTCF sites and unmethylated CGIs are the predominant sites that gain methylation. No functional investigation or follow up analysis are provided.

In conclusion, pending the above mentioned additional QC metrics, the data will be valuable for the community and hence an advance even if the paper itself does not provide major insights.

Extended Figure 1 seems missing

Referee #2 (Remarks to the Author):

The paper "A DNA methylation atlas of normal human cell types" from Loyfer et al presents DNA methylation reference profiles for approximately 40 cell-types, derived from 207 FACS-sorted samples from dissociated healthy tissue, generated with WGBS at 32x. By using WGBS, this atlas represents a significant improvement over a previous DNAm-atlas generated with Illumina beadarrays by Moss et al Nat Commun 2018. In particular, the much higher coverage of WGBS 32x, allows the inference of methylation blocks (contiguous localized regions where DNAm levels are highly correlated), which should be functionally more important than markers identified at the single CpG level (as done by Moss et al). As such, this DNAm-atlas is much better suited for cell-type deconvolution of cfDNAm profiles, which is an extremely important and vibrant application area for personalized medicine. Indeed, the authors provide a beautiful application of their atlas to plasma from Covid-19 patients, discovering that the endothelial cfDNA fraction correlates with disease severity. They also provide an improved characterization of the composition of cfDNA in healthy plasma, which is extremely important for downstream applications. Another significant advantage of this DNAm atlas over the one presented by Moss et al, is that it allows much better characterization of cell-type specific enhancer regions and identification of cell-type specific transcription factors. However, in this regard, a major disappointment is that the authors do not provide us with a complete catalogue of cell-type specific enhancers and TFs, which is a great shame, as I would argue that this is needed for a journal like Nature. The DNAm-atlas resource generates other important insights, notably, that the overwhelming majority of cell-type specific differentially methylated regions are unmethylated in a given cell-type, yet some of the other insights are not really novel. The cell-type deconvolution analyses are promising but lack substantial clarity, appear incomplete, and I have also identified a number of serious concerns which strongly question some of the claims made. Finally, the biggest concern of all, which significantly dampens my enthusiasm for this MS is that the high purity of their 207 samples is not demonstrated. It is essential that a DNAm-atlas should contain high purity (>90%) samples, yet no actual data to support this is provided.

In summary, whilst the presented DNAm-atlas resource has the potential to be of great value to the community, there are some very serious concerns that significantly dampen my enthusiasm for this resource, and indeed I would argue that there is big room for improvement. This MS could be so much better.

Major concerns:

- 1) Purity of the FACS-sorted samples is not demonstrated: By far, one of the biggest concerns with this work is that the purity of the 207 FACS-sorted samples is not demonstrated. I was actually quite shocked by this, because any DNAm-atlas should be based on samples of high purity. There is no mention of the purity in the main text. In Methods there is a vague reference to Ext.Data.Fig.S1 (which seems to be missing from the merged file) and a sentence that states "Purity of live sorted cells was determined by mRNA analysis for key known cell-type specific genes whereas purity of cells that were fixed prior to sorting was determined using previously validated cell-type specific

methylation signals". In the former case, we all know that "specific" markers used in FACS sorting do NOT ensure purity as clearly demonstrated now by a growing number of scRNA-Seq studies (see eg. Ranzoni & Cvejic A Cell Stem Cell 2021 for examples of this in the hematopoietic system). In the latter case, there is no mention of where these validated cell-type specific methylation signals come from, and there is no estimate of the purity of the samples. In summary, the purity of the samples going into the construction of this resource is unclear, and therefore this has to be seen as a huge limitation. Incidentally, let me point out that the observations of replicates clustering together or that of pancreatic cells clustering together does NOT prove high (i.e >90%) purity. You would also see such clustering with relatively impure samples.

2) Definition of cell-type specific markers only suggests 70% purity: In methods, the authors state that they selected cell-type specific unmethylated markers as those with an average DNAm per block < 33%, with average DNAm in the other cell-types >66%. The fact that the authors have used a fairly relaxed threshold of 33% with 32x WGBS suggests to me that their FACS-sorted samples are on average only ~70% pure. Otherwise, the authors should please explain why they did not use more stringent threshold (<10% & >90%).

3) Cell-type specific enhancers and transcription factors (TFs): whilst the observation that cell-type specific unmethylated regions are enriched for enhancers is not entirely novel, one must recognize that the author's resource presents the most conclusive evidence for this. As such, this DNAm-atlas resource could be used to identify novel cell-type specific enhancers and TFs, notably distal enhancers and TF regulatory circuits, as pointed out by the authors at the end of that subsection. However, disappointingly the authors did not develop this further, which is a pity. In fact, since the biggest selling point of this paper is the resource itself, it is paramount that the authors provide a compendium of cell-type specific enhancers and TFs for the 40 cell-types. The authors should provide this data in the form of Supp.tables and provide supporting evidence from gene-expression studies, even scRNA-Seq. I think that this is critically important for consideration in a journal like Nature. This could also indirectly help address the purity issue.

4) In silico simulation model of cfDNA profiles could be more realistic: in Fig.6A the authors present the result of an analysis on in-silico generated mixtures where the authors mixed reads from one cell-type with leukocyte reads, i.e. in effect a 2 cell-type mixture. However, as subsequently shown by the authors, healthy plasma contains non-negligible fractions of hepatocytes and vascular endothelial cells, besides blood cells. In other words, in a disease context, say early detection of lung cancer, we may have more than 2 broad cell-types. Therefore, I feel that the simulation model results presented in Fig.6A is not ideal, as the 2 cell-type scenario is definitely much easier than say a 4 cell-type mixture scenario where one would have endothelial and hepatocyte cells in addition to blood and say lung cancer cells. I would thus urge the authors to consider a more challenging and realistic simulation model.

5) In silico cell-type deconvolution analysis could be overfitted/biased (Fig.6A): A big concern with Fig.6A is that the authors have been extremely unclear about how exactly the data in Fig.6A were generated. In particular, if the authors generated the in-silico mixtures from the same samples used to generate the DNAm reference matrix (i.e. with the 25 cell-type specific markers per cell-type), then the performance will be severely biased & inflated. When I read Methods, the authors seemingly used a training-test set strategy, but fail to clearly explain the exact procedure. The correct procedure would be to select the cell-type specific markers making up the DNAm reference matrix from a training set, and to then build the in-silico mixtures from test set samples that were never used in the selection of cell-type specific markers. However, as mentioned, the authors

description in Methods is so unclear and vague, that this does not inspire much confidence in what the authors have done. If overfitting and inflation is indeed present that could well explain the extraordinarily good correlations seen in Fig.6A.

6) Comparison to Moss et al DNAm-atlas in Fig.6A is potentially biased: In the same vein, the data displayed in Fig.6A and obtained from the Moss et al DNAm-atlas is also shrouded in mystery. In Methods, the authors state that 450k data was “simulated”, but I wonder why any simulation is needed? Surely, on exactly the same WGBS mixtures used to evaluate their own DNAm-atlas, one should be able to collapse DNAm values at the level of Illumina probes to allow cell-type fraction estimation using the DNAm-atlas from Moss et al? However, this analysis would also be biased against Moss et al, since the authors are evaluating the two atlases on the WGBS data generated in this study. To make the comparison fair, the authors should also generate in-silico mixtures using Illumina DNAm data of purified cell-types (there is sufficient data in the public domain, including even Moss et al), and then devise a method to perform cell-type fraction estimation starting out from their WGBS-based DNAm-atlas, and compare results to Moss et al’s DNAm-atlas. If the mixtures are generated using data from Moss et al this may favour the Moss et al DNAm-atlas, but if completely independent Illumina DNAm profiles were used, then this would provide a more unbiased picture. I urge the authors to attempt all of these additional analyses. In any case, the poor performance of the Moss et al DNAm-atlas, as evident from Fig.6A, demands an explanation.

7) Fig.6E-I: That the Epigenomics Roadmap tissue samples are heterogeneous entities is absolutely no surprise. Liver tissue is certainly not just hepatocytes. That lung tissue contains such a high fraction of endothelial + immune cells has also been previously observed (see Zheng SC et al Epigenomics 2018 and EpiSCORE paper-Teschendorff et al Genome Biol.2020). So, the data shown in Fig.6E-I only validates the DNAm-atlas in complex tissues in so far as it predicts substantial heterogeneity, but whether the estimated cell-type fractions are good proxies for those in the actual sample has not been demonstrated. If suitable WGBS is not available, could the authors perhaps assess their DNAm-atlas in Illumina 450k data from epidermis and dermis (Vandiver & Feinberg Genome Med 2014), since skin is a tissue where the relative proportions of keratinocytes, fibroblasts, endothelial cells and immune-cells in the dermis and epidermis are known (see Zhu T, Liu J et al Nat Methods 2022). Or in lung samples, the authors could try to cross-compare their estimates with those using EpiSCORE. In effect, whether the DNAm-atlas presented here could be used for cell-type deconvolution of solid tissues has not been demonstrated.

8) Thresholds and procedures in the definition of cell-type specific markers are not well justified: there are a number of steps in the definition and selection of cell-type specific markers that were not well justified. For instance, blocks with insufficient coverage (number of observations <25) were assigned a value 0.5. So, suppose that for 6 reads covering 4 CpGs each, all are unmethylated, the authors decided that the evidence is stronger for this block to have a DNAm value of 0.5 as opposed to something like 0.1??? I am perplexed by this. You can’t just impute blocks with few observations with a value of 0.5, WHY 0.5??? And indeed, where is the statistical justification for using 25 observations as the threshold? In the next paragraph, the authors describe a procedure for calculating delta-beta between one cell-type and the rest, but I did not understand this. Please clarify- I advise the authors sketch a figure that illustrates the procedure.

9) Impact of age on the segmentation and blocks: I note that the samples derive from a very wide age-range (3 to 83 years). As the authors will know, DNAm patterns change significantly with age, and these changes can be quite stochastic, i.e. the spatial correlative nature of DNAm does get somewhat eroded. Since the segmentation procedure relies on spatially correlated patterns, I

wonder if the authors have a sense of how older samples could alter the segmentation landscape? Another important question to address would be the residual heterogeneity in DNAm levels of the inferred blocks? I presume some blocks may be much more homogenous than others? Some blocks may also contain single CpG outliers? Does this within-block heterogeneity increase with age?

10) The UXM fragment deconvolution algorithm: In the statement “Given an input sample, we compute the U or M read count within each marker as a 1,266x1 vector...” it is terribly unclear where the number 1266 comes from? This number appears from nowhere! Moreover, subsequently, when describing the NNLS procedure, the authors do not define what “b” is! I presume b is the sample profile that needs to be decomposed. The NNLS procedure itself is also odd, in the sense that the authors appear to renormalize the reference DNAm matrix A by the coverage of the input sample, but would it not be much better to normalize b so that its entries are proportions. That would be much better because the entries in the A-matrix are proportions and the coefficients x to be estimated should also be proportions that add to 1. Indeed, both formulations are probably equivalent and lead to the same final answer, but intuitively it makes a lot more sense to normalize the input sample vector b so that all three quantities (A, b and x) are on the same scale (0 to 1).

11) Insensitivity of DNAm reference profiles to genetics and environment is not a novel observation: whilst the authors have used WGBS, the observation that replicates of the same cell-type clustered together irrespective of donor (therefore independent of genetics and factors such as age) is not novel. Indeed, this has already been observed by many previous studies using Illumina arrays (e.g. Reinius LE PLoS One 2012), and it would be extremely surprising if results were different using WGBS. Hence, please tone this down and cite papers like Reinius et al.

12) Methylation patterns record human developmental history: I only partly agree with the author’s interpretation that samples cluster by lineage and not function. Since the purity of their samples is questionable (and very likely purity is only around 70%), that the pancreatic endocrine “cells” cluster with the exocrine acinar/ductal “cells” may have more to do with the fact that the endocrine and exocrine samples are not pure. The author’s endocrine samples probably contain 30% exocrine cells and vice-versa. This could also be a strong reason why these endocrine cells do not cluster closer to neurons. Moreover, the pancreatic endothelial cells cluster more closely with adipocytes, than with the other endothelial cells from other tissue types, which I think has more to do with impurity of their samples? Also, that basal and luminal breast samples cluster together may have more to do with the fact that these samples are not pure (ie the basal samples may contain 30% luminal cells and vice-versa). A pity with this DNAm-atlas is that the authors did not profile resident macrophages from different solid tissue-types. In summary, I think that the authors need to recognize that impurities of their samples can strongly confound their interpretation, so some toning down is advised.

13) Extended Data Fig.5: I only partly agree with the authors that the Epigenomic Roadmap samples (including the primary cell samples) are more contaminated, which may explain the observation depicted in this figure. The author’s interpretation is a little speculative in the absence of clear data proving this. The authors seem to be implying (wrongly so) that their 207 samples are of high >90% purity. Where is the proof of that? Incidentally, that NIH Epigenomics Roadmap primary cell samples could be contaminated was first indicated in the following publication (Zheng SC et al Epigenomics 2018), where it was shown that one of the podocyte samples was not pure.

14) Comparison to other DNAm-atlas approaches: In introduction, and then again in Discussion, it would be good if the authors were to mention, contrast & discuss the recent EpiSCORE and pan-tissue DNAm-atlas papers (Teschendorff et al Genome Biol.2020 & Zhu T, Liu J et al Nat Methods

2022), which provide a complementary strategy to generate reference DNAm profiles via imputation from tissue-specific scRNA-Seq atlases. Clearly, the DNAm-atlas presented in this work has many important advantages e.g. it is applicable to cfDNAm-data, but being less tissue-specific it may be less suitable for cell-type deconvolution of specific solid tissues like liver (cholangiocytes missing) or skin (melanocytes missing).

Minor points:

- a) Fig.1B: the color band for Neuron seems to be misaligned? Please check.
- b) The Dynamic Segmentation Programming algorithm: I have a number of concerns/questions regarding the segmentation into blocks. First, it would appear that the algorithm runs iteratively over increasingly number of sites ranked by genomic position. Does that not mean that the final “optimal segmentation” may be influenced by the initialization? I presume the authors have checked that the optimal segmentation is robust to whether we initialize from one end of the chromosome arm or from the other? Does the segmentation take distance between subsequent CpGs into account?

Referee #3 (Remarks to the Author):

This paper generates a cell type DNA methylation atlas to identify cell type specific loci. They show that these loci are useful for deconvoluting mixtures of cells and shed light on their regulation.

Overall there is little doubt that the authors have generated a resource that will be valuable to the community. The data quality appears to be high and the analyses they have carried out are thorough. While the study does not shed light on new mechanisms, it reinforces previously established roles of cell type specific enhancers.

As a result of the clear utility and high quality of the study I have only minor suggestions:

- 1) While the authors report that the reproducibility of samples is high, there is relatively little data provided on the quality of the WGBS profiles. I would suggest providing more comprehensive plots of coverage and methylation levels across samples in all sequence contexts. For example, previously it was reported that neurons have high non-CG methylation, and it would therefore be of interest to see if this was observed in this study as well.
- 2) I found the discussion of target genes regulated by cell type enhancers to be somewhat confusing. It would be helpful to clarify why the bidirectional zscore approach is appropriate in this context
- 3) The analysis of COVID samples is potentially interesting, but it would be stronger if a quantitative comparison with previously reported results was made.
- 4) The study is performed on HG19, presumably because of the wealth of ChIP data that is available mapped to that build. However, newer builds are now available and it would be useful to consider

using those instead.

5) It would be helpful if the authors provide a brief description of the formats in which the data is made available

Author Rebuttals to Initial Comments:

Detailed response to reviewer comments

We thank the reviewers for their thoughtful comments and overall support of the manuscript. We have addressed all concerns, as detailed in the point by point rebuttal below, and have incorporated the new analyses and clarifications in the revised manuscript and the supplemental information. Most importantly, we have provided detailed information on the purity of samples used to generate the methylome atlas, including new validation, and have performed new analyses of the WGBS data to facilitate studies of genome-wide enhancers and transcription factor binding.

Referee #1 (Remarks to the Author):

Loyfer et al report WGBS data for 207 samples capturing 39 cell types. The average coverage is 30x with the lowest being 5x. The authors then identify methylation blocks (2.8M). Samples of the same type are highly correlated. Unsupervised clustering recreated major lineage trees. Differential methylation analysis finds cell type specific unmethylated blocks enriched for TF binding sites and enhancers and reports a small percentage of regions that gain methylation as expected at canonically unmethylated sites (CTCF sites and CGIs). The unique signatures can be used to deconvolute cell mixtures. This is clearly a massive endeavour and major data generation effort.

Such an extensive set of WGBS data is clearly a valuable resource. The sequencing coverage (average 30x) is standard for small sample numbers but likely limiting for large scale comparisons. Unfortunately, some key numbers are not reported: i) the bisulfite conversion rate determined through CC conversion rates (not spike ins) needs to be provided, ii) the number of CpGs covered at 1x and 10x in each sample. To the above point, how many CpGs are covered at 10x across all 207 samples?

We have now added this information to Table S1. For each of the 207 samples we report the number of sequenced read-pairs, the average coverage (where each pair is counted as one), the percent of CpGs at $\geq 1x$ and at $\geq 10x$ coverage, and the average methylation levels at non-CpG sites (CHH and CHG) and CpG sites.

Briefly, an average of 96% of the CpG sites are covered at $\geq 1x$, 91% at $\geq 10x$ and 80.6% at $\geq 20x$. The average bisulfite conversion rate at non-CpG sites was 98.1% (median 98.8%). Conversely, the average methylation levels at CpG sites was 76.5%.

The current Figures are not very informative, in particular Figures 1-4. Most of these overview panels could be in the supplement and the main text should have better examples that showcase the data and new biology. The text is often redundant and the discussion is largely a repeat summary of the results.

Overall, the data sets have a lot of potential but currently a very limited number of insights are presented and the few selected highlights are not entirely new and not properly discussed in the context of prior literature. For instance, it is well established that cell- and tissue-specific methylation differences are found at enhancers. Nothing of relevance is reported for the hypermethylated regions. It is expected that unmethylated CTCF sites and unmethylated CGIs are the predominant sites that gain methylation. No functional investigation or follow up analysis are provided.

In conclusion, pending the above mentioned additional QC metrics, the data will be valuable for the community and hence an advance even if the paper itself does not provide major insights.

We thank the reviewer. The QC data have been added to the paper. In the revised version we have also added extensive information on the landscape of unmethylated regions, including putative enhancers and TF binding in each cell type. We agree that this is mostly a resource for the community, from which biological insights can be obtained. In addition, the atlas provides a particularly important resource for the practical purpose of determining tissue purity and liquid biopsy applications.

Extended Figure 1 seems missing

Extended Figure 1 has now been revised and purity estimations were added (here and in Extended Figure 2). Full details on cell sorting (e.g. protocols, FACS sorting, and RNA/methylation enrichments) are included in the Supplemental Information.

Referee #2 (Remarks to the Author):

The paper “A DNA methylation atlas of normal human cell types” from Loyfer et al presents DNA methylation reference profiles for approximately 40 cell-types, derived from 207 FACS-sorted samples from dissociated healthy tissue, generated with WGBS at 32x. By using WGBS, this atlas represents a significant improvement over a previous DNAm-atlas generated with Illumina beadarrays by Moss et al Nat Commun 2018. In particular, the much higher coverage of WGBS 32x, allows the inference of methylation blocks (contiguous localized regions where DNAm levels are highly correlated), which should be functionally more important than markers identified at the single CpG level (as done by Moss et al). As such, this DNAm-atlas is much better suited for cell-type deconvolution of cfDNAm profiles, which is an extremely important and vibrant application area for personalized medicine. Indeed, the authors provide a beautiful application of their atlas to plasma from Covid-19 patients, discovering that the endothelial cfDNA fraction correlates with disease severity. They also provide an improved characterization of the composition of cfDNA in healthy plasma, which is extremely important for downstream applications. Another significant advantage of this DNAm atlas over the one presented by Moss et al, is that it allows much better characterization of cell-type specific enhancer regions and identification of cell-type specific transcription factors. However, in this regard, a major disappointment is that the authors do not provide us with a complete catalogue of cell-type specific enhancers and TFs, which is a great shame, as I would argue that this is needed for a journal like Nature. The DNAm-atlas resource generates other important insights, notably, that the overwhelming majority of cell-type specific differentially methylated regions are unmethylated in a given cell-type, yet some of the other insights are not really novel. The cell-type deconvolution analyses are promising but lack substantial clarity, appear incomplete, and I have also identified a number of serious concerns which strongly question some of the claims made. Finally, the biggest concern of all, which significantly dampens my enthusiasm for this MS is that the high purity of their 207 samples is not demonstrated. It is essential that a DNAm-atlas should contain high purity (>90%) samples, yet no actual data to support this is provided.

In summary, whilst the presented DNAm-atlas resource has the potential to be of great value to the community, there are some very serious concerns that significantly dampen my enthusiasm for this resource, and indeed I would argue that there is big room for improvement. This MS could be so much better.

We thank the reviewer for the comprehensive analysis and the appreciation of the potential value of the study. As detailed below, we have now addressed all concerns, including the provision of additional data on enhancers and TF, full details on sample preparation and demonstration of high purity.

Major concerns:

1) Purity of the FACS-sorted samples is not demonstrated: By far, one of the biggest concerns with this work is that the purity of the 207 FACS-sorted samples is not demonstrated. I was actually quite shocked by this, because any DNAm-atlas should be based on samples of high purity. There is no mention of the purity in the main text. In Methods there is a vague reference to Ext.Data.Fig.S1 (which seems to be missing from the merged file) and a sentence that states “Purity of live sorted cells was determined by mRNA analysis for key known cell-type specific genes whereas purity of cells that were fixed prior to sorting was determined using previously validated cell-type specific methylation signals”. In the former case, we all know that “specific” markers used in FACS sorting do NOT ensure purity as clearly demonstrated now by a growing number of scRNA-Seq studies (see eg. Ranzoni & Cvejic A Cell Stem Cell 2021 for examples of this in the hematopoietic system). In the latter case, there is no mention of where these validated cell-type specific methylation signals come from, and there is no estimate of the purity of the samples. In summary, the purity of the samples going into the construction of this resource is unclear, and therefore this has to be seen as a huge limitation. Incidentally, let me point out that the observations of replicates clustering together or that of pancreatic cells clustering together does NOT prove high (i.e >90%) purity. You would also see such clustering with relatively impure samples.

We thank the reviewer for this comment. Maintaining high purity was our top priority, and samples that did not meet our criteria were not included in the human methylation atlas. Yet, we should have demonstrated this point more clearly. Below please find demonstration and quantification of the atlas purity, using several complementary measures. These results are also included in the revised Extended Figures S1 and S2, with additional discussions and results included in the Supplementary Information section.

First, Extended Figure S1 was omitted in error from the original manuscript, and is now included as Supplementary Information and as the revised Extended Figure S1, where we demonstrate the purification process for each cell type. These figures show the FACS gating used for each cell type, as well as enrichment analysis for sorted cells. This was done using RNA expression of marker genes (measured by qPCR in the purified sample compared to the whole tissue, in cases where live sorting was done which permitted RNA extraction), or fragment-level enrichment of uniquely unmethylated DNA fragments, using targeted PCR-sequencing on several amplicons for each cell type. Markers used were established previously, independently of our atlas.

For example, cell sorting of endocrine pancreatic cells is demonstrated below, using immunostaining for glucagon (alpha cells, y-axis, left), C-peptide (beta cells, x-axis), and somatostatin (delta cells, y-axis, center). Note good separation, lack of double positive cells, and absent signal when only fluorescent secondary antibodies were added (right). Note that since the hormones are intracellular, their staining requires fixation and penetration so no RNA can be extracted after the sort.

Second, we use a fragment-level purity analysis. We used 450K data to select an independent set of specifically unmethylated regions in various pancreatic cell types. For each marker, we designed targeted PCR amplicons, covering 4-7 neighboring CpGs, and measured the percent of unmethylated fragments in each cell type. As shown here (also in the revised Extended Figure S1), $\geq 90\%$ of fragments in 3 of 4 acinar samples are unmethylated, compared to less $\leq 5\%$ in all alpha, beta, delta, duct, endothelial cells and blood cells.

AllT-1, allT-2, etc. refer to molecules in which all cytosines but one or two were unmethylated (read as T); such molecules are also clearly derived from the cell type of interest and cannot derive from potential contaminating cell types, where they constitute a negligible minority. The top left figure shows minimal contamination of the acinar cell samples (hence the near 100% values) as well as minimal acinar contamination in other pancreatic cell types.

Similar analyses, using markers specifically unmethylated in alpha, beta, delta, duct and endothelial cells, show purity of ~90% for all these cell types. Vascular endothelial cells showed a more heterogeneous methylation pattern, but marker analysis argues against contamination from other pancreatic cell types (since the endothelial preparation does not contain the hallmark “fully unmethylated” version of any other pancreatic cell type).

These estimations were done using targeted PCR. We also examined these exact regions using the WGBS atlas samples, and found almost identical results (below, in green). These results are also included in the revised Extended Figure S1.

We note that this is an under-estimation of the purity, since a genomic region could show some methylation heterogeneity even in a pure cell type. Below we show similar analysis of other regions (markers), suggesting even higher purity in pancreatic cells.

Third, we reason that if indeed a sample is contaminated by one or more cell types, we should see DNA from that other type across the entire genome at a proportion that reflects the contamination. We can therefore focus on any locus, and calculate the percent of fragments that are unmethylated in each cell type, obtaining a lower bound of the purity. We demonstrate below the purity by focusing on one of the top 25 markers, which were selected for being differentially methylated.

Pancreas cell type analysis, suggests ~90% purity for all cell types (except for pancreatic vascular endothelial cells that are either more heterogenous or present at a lower purity of 70-80%):

Lung cell types (pleura and alveolar epithelial cells, bronchial epithelium, interstitial and alveolar macrophages, alveolar endothelial cells, and smooth muscle cells), showing high purity. Leukocyte (granulocyte) markers are shown here and in other tissues to demonstrate lack of blood contamination in any isolated cell type.

Heart cell types, including cardiomyocytes, cardiac fibroblasts, endothelial cells, and smooth muscle cells (again, with ~90% purity):

Liver cell types, including hepatocytes, endothelial cells, and macrophages:

Breast cell types, including basal, luminal epithelial cells, and adipocytes:

Finally, **gastrointestinal tract** cells, including epithelial cells from the esophagus, stomach, small intestine and colon, as well as colon fibroblasts, and colon macrophages.

These results demonstrate purity of ~90% or more for almost all cell types included in the atlas. Results are included in the revised manuscript as Extended Figure S2. A full comparison of all 953 markers in all 207 samples is also included, as Extended Figure S5.

Fourth, following the Reviewer's advice, we demonstrate our purity using markers near EpiSCORE marker genes (Zhu et al, Nature Methods, 2022). While the EpiSCORE values for some cell types were indeed at ~70%, as Reviewer 2 had suspected, we believe that these estimations are mainly due to the use of array-based metrics on WGBS data. Such an approach suffers from several caveats. EpiSCORE is based on average methylation levels at a few CpG sites. Yet, WGBS data has limited depth compared to the effective number of molecules measured by DNA methylation arrays. Consequently, WGBS estimations at single CpG methylation are often crude. In addition, different technological platforms have different biases. Finally, a fragment-level NGS approach generally benefits from CpG-rich regions, where one sequenced fragment reports the methylation state of multiple neighboring CpGs. Conversely, EpiSCORE is based on promoters whose methylation levels are anti-correlated with expression levels. Typically, these are non-CpG-island promoters, hence with fewer CpGs that are further spaced.

Given these considerations we examined the marker genes used by EpiSCORE (for pancreatic cell types, where Reviewer 2 estimated limited purity), but quantified sample

purity using a fragment-level analysis at differentially methylated regions, overlapping the original EpiSCORE CpGs or in their proximity.

Two acinar marker genes used by EpiSCORE are CTRB2 and CELA3B. For CTRB2, a fragment-level analysis of the same genomic region as EpiSCORE yields a purity estimation of ~95%:

On the left, shown in purple are methylation levels across all 207 samples. The block we analyze (chr16:75240963-75241306, 12 CpGs) is marked in red at pancreatic endothelial cells (top) and alpha, beta, delta, acinar and duct (bottom). EpiSCORE CpGs are denoted by asterisks at the bottom. On the top right plot, Green bars show the percent of unmethylated sequenced fragments (≥ 4 CpGs). A purity of ~95% can be

seen for all four acinar samples. Similarly, orange bars show the average methylation levels at the same CpGs, suggesting that some stochasticity in single CpG methylation is present, highlighting the sensitivity of a multi-CpG fragment-level analysis.

Similar results are shown on a differentially methylated regions near CELA3B, another EpiSCORE marker gene for acinar cells, with ~90% of ≥ 4 -CpG fragments unmethylated in acinar samples, compared to near zero percent in other pancreatic samples. Note that given the absence of markers from other cell types in the acinar preps (as shown above), the likely interpretation of 90% demethylation is heterogeneity, i.e. the acinar preps are probably of even higher purity.

Two endocrine markers show a similar purity. Indeed, a genomic region (chr18:29172689-29172840) near the TTR gene promoter is unmethylated in ~90% of fragments in islet cells (alpha, beta, and delta) cells, compared to near zero percent in acinar, duct and endothelial cells:

Finally, FXJD2 is used as a marker gene for duct and endocrine cells, and fragment-level analysis of a genomic regions (chr11:117690879-117690986) shows unmethylated fragments in these cells, but not in acinar and endothelial cells.

These results are an under-estimate of purity, as even a 100% sample would show variation in its fragment-level methylation, e.g. due to epigenetic heterogeneity as noted above, stochastic noise, or age-related changes. These findings are now included in the Supplementary Information section.

Finally, we undertook a comprehensive computational approach (following Reviewer 2 suggestion), and used a **leave-one-out cross-validation** method to segment the genome without the held-out sample, find markers for all cell types based on a reduced atlas, and analyze the held-out sample with the UXM deconvolution algorithm. The estimated contributions of the held-out sample can be used as a rough estimate to its purity, as well as to the homogeneity of the atlas, and the robustness of segmentation and marker selection processes. As we now show in Table S11, the average purity of cell types with ≥ 3 replicates is estimated at 89%, with **median purity of 90%**. 94% of these cell types (30/32) showed “purity” $\geq 80\%$ in this harsh leave-one-out process, where both the segmentation and the marker selection are based on a majorly reduced set of samples.

We thank the reviewer for their detailed suggestions and hope these updated results demonstrate the overall high purity of the atlas samples.

2) Definition of cell-type specific markers only suggests 70% purity: In methods, the authors state that they selected cell-type specific unmethylated markers as those with an average DNAm per block $< 33\%$, with average DNAm in the other cell-types $> 66\%$. The fact that the authors have used a fairly relaxed threshold of 33% with 32x WGBS suggests to me that their FACS-sorted samples are on average only ~70% pure. Otherwise, the authors should please explain why they did not use more stringent threshold ($< 10\%$ & $> 90\%$).

The threshold of 33% and 66% were only used to identify a rough set of differentially methylated regions. The top 25 markers per cell type (1232 in total) are in fact much more diverged as Reviewer 2 predicted, with an average methylation of 13% within their target cell types, and 91% elsewhere, as shown in Supplemental Table S4, columns J and K, and consistent with a purity estimation of ~90% or higher.

Fragment-level analysis of these 1,232 markers shows even higher estimations. When examining fragments of ≥ 4 CpGs, 86% of the fragments in a given cell type are unmethylated in that cell type, compared to only 3% of fragments in other cell types. As noted above, these could be under-estimates due to true epigenomic heterogeneity or biological stochasticity.

3) Cell-type specific enhancers and transcription factors (TFs): whilst the observation that cell-type specific unmethylated regions are enriched for enhancers is not entirely novel, one must recognize that the author's resource presents the most conclusive evidence for this. As such, this DNAm-atlas resource could be used to identify novel cell-type specific enhancers and TFs, notably distal enhancers and TF regulatory circuits, as pointed out by the authors at the end of that subsection. However, disappointingly the authors did not develop this further, which is a pity. In fact, since the biggest selling point of this paper is the resource itself, it is paramount that the authors provide a compendium of cell-type specific enhancers and TFs for the 40 cell-types. The authors should provide this data in the form of Supp.tables and provide supporting evidence from gene-expression studies, even scRNA-Seq. I think that this is critically important for consideration in a journal like Nature. This could also indirectly help address the purity issue.

We thank the reviewer for this suggestion. The revised version of the manuscript contains the top 25 markers for each cell type, which are most specific and suitable for plasma and tissue deconvolution; the top 250 markers for each cell type, which were used for chromatin enrichments shown in Figure 4A; and the top 1,000 markers which we used for transcription factor motif analysis, shown in Figure 4B and Table S6A. These sets are included in the revised manuscript either in Table S4, or as supplemental datasets. As we have previously shown, these regions have enhancer characteristics (H3K27ac, H3K4me1, DNA accessibility, chromHMM enhancer annotations, and enrichment for relevant transcription factors binding sites), but they are mostly unique to one cell type, namely unmethylated in one cell type, and methylated in others.

For a more complete cell type-specific enhancer catalog, we have now taken another approach where each cell type is analyzed independently. We devised a fragment-level analysis, by which the entire genome is scanned (for every cell type), and genomic regions (blocks of ≥ 4 CpGs) with $\geq 85\%$ of unmethylated fragments are identified. For each of the 39 cell types we analyzed, this includes an average of 44,957 unmethylated genomic regions, which are now included as a supplemental dataset in the revised submission.

We annotated these regions with external data, including their genomic annotation, their neighboring gene, an overlap with TSS regions (up to 1Kb), overlap with CpG islands, their chromHMM annotation in the relevant cells, as well as overlap with chromatic ChIP-seq peaks of H3K4me3, H3K27ac, H3K4me1, H3K27me3 in matching cell types or tissues, where available. We also tested each region for an overlap with a CTCF

binding site and CTCF in vivo binding (ChIP-seq peak) in matching conditions. Taken together, these annotations will allow the community to study, across multiple cell types, a large dataset of unmethylated regions that could be further annotated as promoters, putative enhancers, CpG islands, CTCF sites, etc.

Finally we selected for the 32 cell types for which relevant chromatin data is available, the unmethylated regions that are marked by H3K27ac but not by H3K4me3 (namely, active regulatory regions that are not promoters). This yielded an average of ~5,600 genomic regions per cell type. Intriguingly, over half of these regions were indeed annotated by chromHMM as Enhancer regions. We then used JASPAR to identify putative transcription factor binding sites, which are listed for each region.

We also analyzed these regions (the full set of unmethylated regions, as well as the reduced set of unmethylation regions overlapping H3K27ac but not H3K4me3 peaks) for transcription factor motif enrichment (using HOMER), typically identifying the same transcription factors found at the top 1,000 unique enhancers (Figure 4B, Table S6A), but with generally much more significant p-values.

All these cell type-specific sets of unmethylated regions and putative enhancers, as well as annotations and transcription factor binding sites and motif enrichments, are now included as supplemental datasets and in Tables S6B and S6C.

4) *In silico* simulation model of cfDNA profiles could be more realistic: in Fig.6A the authors present the result of an analysis on *in-silico* generated mixtures where the authors mixed reads from one cell-type with leukocyte reads, i.e. in effect a 2 cell-type mixture. However, as subsequently shown by the authors, healthy plasma contains non-negligible fractions of hepatocytes and vascular endothelial cells, besides blood cells. In other words, in a disease context, say early detection of lung cancer, we may have more than 2 broad cell-types. Therefore, I feel that the simulation model results presented in Fig.6A is not ideal, as the 2 cell-type scenario is definitely much easier than say a 4 cell-type mixture scenario where one would have endothelial and hepatocyte cells in addition to blood and say lung cancer cells. I would thus urge the authors to consider a more challenging and realistic simulation model.

We have generated more challenging 4-way mixing as suggested. For this, we first prepared a simulated “plasma” mixture, composed of 90% reads from blood, 7.5% of endothelial cell reads, and 2.5% from hepatocytes. We then mixed reads from either cardiomyocytes, bladder, breast, or erythrocyte progenitor cells, into this mixture, at varying proportions, from 0% through 10%.

In agreement with Reviewer 2's comment #5 below, this was done in a **leave-one-out cross-validation** manner. For example, a cardiomyocyte mix would use a new segmentation of the genome, as well as a new set of cell type-specific markers, based on a reduced set of 204 samples. These include the original 207 samples, minus three held-out samples: the endothelial, hepatocyte, and cardiomyocyte that are mixed in with blood. The mixed sample is then deconvolved using the UXM algorithm, and the predicted cardiomyocyte proportion is reported and compared to the actual one.

As these updated results demonstrate, the methylation atlas with 25 markers per cell type has an accuracy of ~0.3%, even in the context of more complex mixtures that simulate plasma cfDNA.

Intriguingly, the different sets of cell type-specific markers, obtained from the reduced atlases after the removal of one endothelial, one hepatocyte, and one additional sample from each run, are 94% identical to the original set of top 25 markers, obtained from the full set of 207 samples, suggesting that the set of cell type-specific markers is quite robust. These *in silico* 4-way results are now reported as Extended Figure S7, in the revised manuscript.

5) *In silico* cell-type deconvolution analysis could be overfitted/biased (Fig.6A): A big concern with Fig.6A is that the authors have been extremely unclear about how exactly the data in Fig.6A were generated. In particular, if the authors generated the *in-silico* mixtures from the same samples used to generate the DNAm reference matrix (i.e. with the 25 cell-type specific markers per cell-type), then the performance will be severely biased & inflated. When I read Methods, the authors seemingly used a training-test set strategy, but fail to clearly explain the exact procedure. The correct procedure would be to select the cell-type specific markers making up the DNAm reference matrix from a training set, and to then build the *in-silico* mixtures from test set samples that were never used in the selection of cell-type specific markers. However, as mentioned, the authors description in Methods is so unclear and vague, that this does not inspire much

confidence in what the authors have done. If overfitting and inflation is indeed present that could well explain the extraordinarily good correlations seen in Fig.6A.

We thank the reviewer for the comment. Indeed, the results previously presented in Figure 6A are based on the full set of 207 samples for segmentation and marker identification. We did not expect the segmentation or selected markers to alter when one sample is removed, and preferred to use one atlas for all samples.

Following the reviewer's concern, we have now repeated this analysis using a **leave-one-out** approach (similar to the one described above, for the 4-way in silico mixing). In brief, for each in silico mixing run, we removed one sample from the atlas (e.g. a cardiomyocyte sample), rerun the segmentation and marker selection procedure on the remaining 206 samples to obtain a new set of blocks and markers, mixed the held-out sample with blood data (external to the atlas), and analyze the mix using the new "reduced" markers. This resulted with similar plots, shown here and in the revised Figure 6A:

Intriguingly, the removal of a single sample hardly affected the genome-wide segmentation and marker selection, with an average of 97.3% of markers unchanged, across 40 leave-one-out runs (6 cardiomyocytes, 5 Bladder, 4 Breast basal and 3 luminal, 19 endothelial samples, and 3 Eryth. Prog), compared to the full set of 1,232 top 25 markers obtained by the full set of 207 samples (Table S4).

6) Comparison to Moss et al DNAm-atlas in Fig.6A is potentially biased: In the same vein, the data displayed in Fig.6A and obtained from the Moss et al DNAm-atlas is also shrouded in mystery. In Methods, the authors state that 450k data was "simulated", but I wonder why any simulation is needed? Surely, on exactly the same WGBS mixtures used to evaluate their own DNAm-atlas, one should be able to collapse DNAm values at the level of Illumina probes to allow cell-type fraction estimation using the DNAm-atlas from Moss et al? However, this analysis would also be biased against Moss et al, since the authors are evaluating the two atlases on the WGBS data generated in this study. To make the comparison fair, the authors should also generate in-silico mixtures using Illumina DNAm data of purified cell-types (there is sufficient data in the public domain,

including even Moss et al), and then devise a method to perform cell-type fraction estimation starting out from their WGBS-based DNAm-atlas, and compare results to Moss et al's DNAm-atlas. If the mixtures are generated using data from Moss et al this may favour the Moss et al DNAm-atlas, but if completely independent Illumina DNAm profiles were used, then this would provide a more unbiased picture. I urge the authors to attempt all of these additional analyses. In any case, the poor performance of the Moss et al DNAm-atlas, as evident from Fig.6A, demands an explanation.

Figure 6A aimed to quantify the accuracy of fragment-level analysis of sequenced data using the described atlas and algorithm, and compare these to deconvolution using the array-based atlas of Moss et al. For this, we used the same mixes (of WGBS sequenced data), and computed the average methylation at 450K/EPIC CpG sites. We then ran the meth_atlas program from Moss et al. on these ~8000 CpG sites, to deconvolute the methylation level in the mixed data using an array-based atlas.

Indeed, the results presented here by the array-like data (and algorithm) are consistent with ~1% sensitivity we estimated in the original Moss et al paper. These were compared to analyzing the same set of mixed reads using the fragment-level UXM approach.

Our atlas and deconvolution algorithm are relevant only to NGS data, and cannot be applied to array-based mixes. These limitations are now discussed (in the Results and Discussion sections). We have also rephrased the Methods section for clarity.

7) Fig.6E-I: That the Epigenomics Roadmap tissue samples are heterogeneous entities is absolutely no surprise. Liver tissue is certainly not just hepatocytes. That lung tissue contains such a high fraction of endothelial + immune cells has also been previously observed (see Zheng SC et al Epigenomics 2018 and EpiSCORE paper-Teschendorff et al Genome Biol.2020). So, the data shown in Fig.6E-I only validates the DNAm-atlas in complex tissues in so far as it predicts substantial heterogeneity, but whether the estimated cell-type fractions are good proxies for those in the actual sample has not been demonstrated. If suitable WGBS is not available, could the authors perhaps assess their DNAm-atlas in Illumina 450k data from epidermis and dermis (Vandiver & Feinberg Genome Med 2014), since skin is a tissue where the relative proportions of keratinocytes, fibroblasts, endothelial cells and immune-cells in the dermis and epidermis are known (see Zhu T, Liu J et al Nat Methods 2022). Or in lung samples, the authors could try to cross-compare their estimates with those using EpiSCORE. In effect, whether the DNAm-atlas presented here could be used for cell-type deconvolution of solid tissues has not been demonstrated.

We thank the reviewer for this suggestion. As discussed above, the atlas and the UXM algorithm presented here are designed for sequencing-based data, and cannot be directly applied to array data. We have now calculated the average methylation (per CpG) in Roadmap samples, and used EpiSCORE for deconvolution.

Loyfer et al. WGBS atlas

EpiSCORE

While the sequencing depth of WGBS samples might hinder this approach, the results are quite interesting, as shown below. In the **Liver** samples (n=3), both algorithms predicted similar compositions, with an estimate of 57% hepatocytes according to our atlas, compared to 40% hepatocytes and 20% cholangiocytes in EpiSCORE. **Lung samples** (n=4) were predicted to contain 22% epithelium, 34% endothelium, and 39% blood according to the WGBS atlas, compared to 7% epithelial cells, 11% stroma, 30%

endothelium, and 48% blood according to EpiSCORE. The **pancreatic islets** sample was predicted to be composed of 47% beta, and 24% alpha, 17% duct and 8% acinar by the WGBS atlas, compared to 83% endocrine (i.e. beta+alpha, 71% per WGBS), 10% acinar, and 5% duct by EpiSCORE. In the **Heart Ventricle** samples (n=6), we estimated 39% endothelium, 35% cardiomyocytes, and 17% fibroblasts, whereas EpiSCORE analysis predicted 24% cardiomyocytes, 12% fibroblasts, and 55% smooth muscle cells. Finally, **colon** samples (n=4) were predicted by our atlas to contain 47% epithelium, 23% fibroblasts, 21% blood, compared to 30% epithelium, 10% fibroblasts, 23% blood, and 24% endothelium, according to EpiSCORE.

8) Thresholds and procedures in the definition of cell-type specific markers are not well justified: there are a number of steps in the definition and selection of cell-type specific markers that were not well justified. For instance, blocks with insufficient coverage (number of observations <25) were assigned a value 0.5. So, suppose that for 6 reads covering 4 CpGs each, all are unmethylated, the authors decided that the evidence is stronger for this block to have a DNAm value of 0.5 as opposed to something like 0.1??? I am perplexed by this. You can't just impute blocks with few observations with a value of 0.5, WHY 0.5??? And indeed, where is the statistical justification for using 25 observations as the threshold?

The minimal value of 25 observations was selected to reflect a tradeoff between the minimal read coverage across samples in each genomic region, and the estimation error in DNA methylation, per block. A more stringent threshold would allow accurate estimations of the methylation level in each sample/block, but will suffer from more missing values. A lower threshold would have almost no missing values, but poor estimations in the case of very few reads.

In the case of Bernoulli random variables, the standard error of a sample mean is $\sqrt{p \cdot q / n}$, with p being the probability of CpG methylation, $q=1-p$, and n is the number of observations. For most unmethylated markers, p in the range of [5%,15%], thus a value of $n=25$ yields an estimation error below 10%, which should suffice for choosing highly differentiated markers. While a higher threshold would allow a more accurate estimation, it would also introduce more missing values (or 0.5's, discussed below), given the sequencing depth of the atlas.

The threshold itself is quite robust. Following the reviewer's request, we repeated the marker selection with thresholds of $n=10$ and $n=40$ observations per block. The resulting sets of markers are mostly unchanged relatively, with an overlap of 97% for $n=10$ and 94% for $n=40$, compared to the initial set of markers at $n=25$.

Regarding the default value of “0.5” used to complete values of insufficient depth. This should have been more carefully justified, and we hope the revised text is clearer. 0.5 was selected in a practical programming decision, to be a default value that is neither methylated nor unmethylated. In the context of finding markers, the program seeks for the regions that are mostly unmethylated in the target group, and mostly methylated in the remaining samples (or vice versa). So replacing N/A’s with 0.5 acts as an opposing force against the selection of loci with low coverage.

Similar results could be achieved by ignoring N/A’s when comparing the differential methylation values, but would require yet another threshold to limit the allowed number of missing values in a block (say, up to 10% of the samples in either the target or background groups). This option results in quite a similar set of markers (90% overlap) with no effect on the downstream analysis. Nonetheless, we prefer using the softer “0.5” version, where each missing value is penalized, compared to a binary threshold on the maximal number of N/As allowed, which may change between different cell types. We have now rephrased the methods section to reflect these considerations.

In the next paragraph, the authors describe a procedure for calculating delta-beta between one cell-type and the rest, but I did not understand this. Please clarify- I advise the authors sketch a figure that illustrates the procedure.

We have now included a detailed description of the marker selection process in the Supplemental Information. Briefly, the average methylation value is calculated for each block (≥ 5 CpGs) in every sample, or set as 0.5 if < 25 observations exist. We then compute the “soft margin” between the target samples and the background samples. Ideally, we would consider the highest (most methylated) value in the target samples, and the lowest (least methylated) value in the background samples, and rank the blocks accordingly, then selecting those with the maximal difference. For example, at the heart cardiomyocytes marker shown in figure 3C (chr17:45289451-45289570), all cardiomyocyte samples would have an average methylation $\leq 7\%$, whereas the remaining 200 samples are at 80% methylation or above. So this min/max margin is demonstrated by vertical solid lines.

To allow some flexibility, we allow single outliers using percentiles. So instead of comparing the min/max values, we compare the lower 25% percentile in the small target group (namely, the 2nd “worst” cardiomyocyte sample, which is at 2% methylation), compared to the 97.5% in the large background group (namely, the 5th “worst” sample, which is at 90%). These are plotted above using dotted lines.

In that sense, replacing N/A's values with 0.5 is acceptable in up to one or two in the target group (or not at all, for groups with $n \leq 3$), and up to ~5 samples in the control groups, after which the effective margin is dramatically reduced. As in the case of a colon epithelial cells marker at chr1:167870100-167870517, in which one of the cardiomyocyte samples contains only 23 observations (which are all methylated), and an imputed value of 0.5 is set.

This is now described in the Supplemental Information section.

9) Impact of age on the segmentation and blocks: I note that the samples derive from a very wide age-range (3 to 83 years). As the authors will know, DNAm patterns change significantly with age, and these changes can be quite stochastic, i.e. the spatial

correlative nature of DNAm does get somewhat eroded. Since the segmentation procedure relies on spatially correlated patterns, I wonder if the authors have a sense of how older samples could alter the segmentation landscape? Another important question to address would be the residual heterogeneity in DNAm levels of the inferred blocks? I presume some blocks may be much more homogenous than others? Some blocks may also contain single CpG outliers? Does this within-block heterogeneity increase with age?

These are fascinating questions regarding aging and tissue-specific methylation, that deserve in-depth investigation, facilitated by the atlas as a resource. To begin addressing this issue, we first compared our markers with age-predicted CpG sites (Horvath, 2013; Hannum et al, 2013; Horvath et al, 2018). We found no overlap between these CpGs and our set of top 1,232 markers.

Regarding segmentation, we divided the 207 samples into 98 samples of age ≤ 50 , and 109 samples above 50, and ran the segmentation process on each set. This had a small effect on the segmentation, as 91% and 94% of the block boundaries were rediscovered again.

Finally, we tested the residual heterogeneity at a fragment-level, by calculating the proportion of “X” fragments in each sample. These are fragments of ≥ 4 CpGs, whose per-read average methylation is between 15% and 85% (i.e. show a combination of methylated and unmethylated CpGs on the same molecule). For each of the 1,232 top 25 markers, we calculated the Spearman correlation coefficient between age and the X proportion across samples. None show a significant increase ($FDR \leq 1e-2$) over age. Further investigations of age-related DNA changes (including fragment-level analysis) are left for future studies.

10) The UXM fragment deconvolution algorithm: In the statement “Given an input sample, we compute the U or M read count within each marker as a $1,266 \times 1$ vector...” it is terribly unclear where the number 1266 comes from? This number appears from nowhere! Moreover, subsequently, when describing the NNLS procedure, the authors do not define what “b” is! I presume b is the sample profile that needs to be decomposed. The NNLS procedure itself is also odd, in the sense that the authors appear to renormalize the reference DNAm matrix A by the coverage of the input sample, but would it not be much better to normalize b so that its entries are proportions. That would be much better because the entries in the A-matrix are proportions and the coefficients x to be estimated should also be proportions that add to 1. Indeed, both formulations are probably equivalent and lead to the same final answer, but intuitively it makes a lot more sense to normalize the input sample vector b so

that all three quantities (A, b and x) are on the same scale (0 to 1).

We apologize for the unclear description and provide a rephrased explanation of the algorithm in the Methods and Supplementary Information sections. This model is a multiple linear regression problem, similar to NNLS, but the numeric values represent the proportion of unmethylated (U) fragments, in each loci, in each cell type, instead of methylation values. The number of features should have been 1,232 for a full set of top 25 markers, with 25 additional megakaryocyte markers that were included in plasma analysis.

As for the coverage normalization: the two formulations are not exactly equivalent, as in the [0-1] case all markers are assigned the same weight (importance), whereas in the coverage-normalized case, markers (=features) with more reads in the input sample “count more”, as to reflect their depth. So they contribute more to the RMSE score we minimize, and therefore are more important. This is described in depth in the revised Supplementary Information.

11) Insensitivity of DNAm reference profiles to genetics and environment is not a novel observation: whilst the authors have used WGBS, the observation that replicates of the same cell-type clustered together irrespective of donor (therefore independent of genetics and factors such as age) is not novel. Indeed, this has already been observed by many previous studies using Illumina arrays (e.g. Reinus LE PLoS One 2012), and it would be extremely surprising if results were different using WGBS. Hence, please tone this down and cite papers like Reinus et al.

We have now rephrased this paragraph and cite Reinus et al (PMID 22848472). We note that even though replicates of the same cell type cluster together, the extent of variation caused by environment (even if it does not reach the scale of disrupting clustering) is an important quantity, which can be assessed only when pure samples are available, as with the atlas.

12) Methylation patterns record human developmental history: I only partly agree with the author’s interpretation that samples cluster by lineage and not function. Since the purity of their samples is questionable (and very likely purity is only around 70%), that the pancreatic endocrine “cells” cluster with the exocrine acinar/ductal “cells” may have more to do with the fact that the endocrine and exocrine samples are not pure. The author’s endocrine samples probably contain 30% exocrine cells and vice-versa. This could also be a strong reason why these endocrine cells do not cluster closer to neurons. Moreover, the pancreatic endothelial cells cluster more closely with adipocytes, than with the other endothelial cells from other tissues types, which I think

has more to do with impurity of their samples? Also, that basal and luminal breast samples cluster together may have more to do with the fact that these samples are not pure (ie the basal samples may contain 30% luminal cells and vice-versa). A pity with this DNAm-atlas is that the authors did not profile resident macrophages from different solid tissue-types. In summary, I think that the authors need to recognize that impurities of their samples can strongly confound their interpretation, so some toning down is advised.

We hope the questions of purity were resolved above to the satisfaction of the reviewer. Specifically, we demonstrate purity of 90% or more for all pancreatic cell types, as well as lung, heart, liver, breast and the GI tract, with no consistent contaminations observed. We have toned down this paragraph (and others). This reinforces the validity of the similarity between endocrine and acinar cells and additional comparisons. We note that even though adipocytes and vascular endothelial cell preparations may not have perfect purity, these cell types do share developmental origins. We did collect tissue-specific macrophages, determined their methylomes and found that they clustered together (as shown in the fanning dendrogram). However, we felt that in-depth analysis of macrophage lineages (which are likely complex) deserves a separate study, with additional experimental data.

13) Extended Data Fig.5: I only partly agree with the authors that the Epigenomic Roadmap samples (including the primary cell samples) are more contaminated, which may explain the observation depicted in this figure. The author's interpretation is a little speculative in the absence of clear data proving this. The authors seem to be implying (wrongly so) that their 207 samples are of high >90% purity. Where is the proof of that? Incidentally, that NIH Epigenomics Roadmap primary cell samples could be contaminated was first indicated in the following publication (Zheng SC et al Epigenomics 2018), where it was shown that one of the podocyte samples was not pure.

We have rephrased and toned down this section, and cited Zheng et al (PMID 29693419).

14) Comparison to other DNAm-atlas approaches: In introduction, and then again in Discussion, it would be good if the authors were to mention, contrast & discuss the recent EpiSCORE and pan-tissue DNAm-atlas papers (Teschendorff et al Genome Biol.2020 & Zhu T, Liu J et al Nat Methods 2022), which provide a complementary strategy to generate reference DNAm profiles via imputation from tissue-specific scRNA-Seq atlases. Clearly, the DNAm-atlas presented in this work has many important advantages e.g. it is applicable to cfDNAm-data, but being less tissue-specific it may be

less suitable for cell-type deconvolution of specific solid tissues like liver (cholangiocytes missing) or skin (melanocytes missing).

Indeed single-cell RNA-seq data offer a compelling complementary strategy for marker selection. EpiSCORE (PMID 32883324) and the recently published pan-tissue atlas (PMID 35277705) are now discussed in the Introduction and Discussion sections. In addition, as noted in the paper the atlas is open for further expansion to missing cell types as mentioned by the reviewer, which would further increase its utility.

Minor points:

a) Fig.1B: the color band for Neuron seems to be misaligned? Please check.

Figure 1B shows a genomic region with differentially methylated regions at various cell types. Yet these regions were not specific enough to rank among the “top 25” set of 1,232 markers. The green bar is correctly positioned, and highlights a region where neurons (top 10 rows) are unmethylated whereas other cell types (including oligodendrocytes, the following 4 tracks) are fully methylated (except endocrine pancreas cell types, which are not fully methylated).

b) The Dynamic Segmentation Programming algorithm: I have a number of concerns/questions regarding the segmentation into blocks. First, it would appear that the algorithm runs iteratively over increasingly number of sites ranked by genomic position. Does that not mean that the final “optimal segmentation” may be influenced by the initialization? I presume the authors have checked that the optimal segmentation is robust to whether we initialize from one end of the chromosome arm or from the other? Does the segmentation take distance between subsequent CpGs into account?

We now include a more informative description above Dynamic Programming algorithms (e.g. HMM), and the assumptions under which they find the optimal segmentation. But briefly, yes, the algorithm is not influenced by the initialization or the relative position within each chromosome. It does so by breaking the chromosome-level segmentation problem into smaller problems. For example, one could solve the segmentation problem of the entire chromosome (positions 1 through n) using previously calculated solutions of all prefixes [1,k]. Let us imagine the optimal segmentation, and now remove the very last segment [k,n]. We are then left with all previous segments, which are the **optimal** segmentation of the prefix [1,k-1]. So at every position i, we basically scan back to position j, and consider whether the optimal segmentation up to the current point [1,i] can be achieved by taking the optimal segmentation [1,j], and adding the remaining position as a new segment [j+1,i]. By restricting the maximal size of a block (5Kb) we can speed-up and limit the running

speed of the algorithm. A typical run of the algorithm across 207 samples takes only a few minutes.

The linear distance (in bp) between subsequent CpGs is indeed ignored, other than a strict threshold of 5Kb for a block length. We also assume independence between neighboring blocks, meaning that the score of the segment [i,j] is independent of how the previous positions are segmented. Our algorithm also ignores fragment-level information, and counts for each CpG how many times it was methylated or not, regardless of other CpGs on the same molecules. We should note that we have also devised a fragment-level segmentation algorithm, which obtained very similar segmentations, and is left outside the scope of this manuscript, for simplicity.

Referee #3 (Remarks to the Author):

This paper generates a cell type DNA methylation atlas to identify cell type specific loci. They show that these loci are useful for deconvoluting mixtures of cells and shed light on their regulation.

Overall there is little doubt that the authors have generated a resource that will be valuable to the community. The data quality appears to be high and the analyses they have carried out are thorough. While the study does not shed light on new mechanisms, it reinforces previously established roles of cell type specific enhancers.

As a result of the clear utility and high quality of the study I have only minor suggestions:

1) While the authors report that the reproducibility of samples is high, there is relatively little data provided on the quality of the WGBS profiles. I would suggest providing more comprehensive plots of coverage and methylation levels across samples in all sequence contexts. For example, previously it was reported that neurons have high non-CG methylation, and it would therefore be of interest to see if this was observed in this study as well.

We thank the reviewer for their kind words and the list of improvements to the manuscript. We have now added detailed QC information in the revised Table S1, including sequencing depth information and conversion rates.

As for the non-CpG methylation, it is indeed slightly higher in neuron samples (2%-7.5%), compared to all other samples where these values vary between 1%-4%, on

par with other WGBS datasets that typically vary between 0.5%-4%. We now report these numbers, but leave the study for non-CpG methylation for future research.

2) I found the discussion of target genes regulated by cell type enhancers to be somewhat confusing. It would be helpful to clarify why the bidirectional zscore approach is appropriate in this context

We thank the reviewer for this comment, and have rewritten this section. The term “bidirectional” was perhaps misleading, referring to iterative Z-score calculations row-wise and column-wise. We are interested in neighboring genes, where (1) the gene is expressed in the target cell type more than in other cell types (row-wise Z-score), and (2) that enrichment is higher for that gene, compared to other genes (column-wise Z-score). So the rationale is that putative cell type-specific enhancers would be associated with genes whose expression is elevated.

In addition to these (somewhat experimental) gene associations, we also associated each marker region in Table S4 and in the supplementary datasets with the nearest gene, and marked whether the marker is exonic, intronic, intergenic, or at the gene promoter. These annotations are based on HOMER’s annotatePeaks.pl function.

3) The analysis of COVID samples is potentially interesting, but it would be stronger if a quantitative comparison with previously reported results was made.

We have now included in the Supplementary Information section a scatterplot comparing the deconvolution results of Cheng et al. (PMID 33521749) and ours, for cell types assessed in both studies.

Overall, as can be seen below, the agreement is quite high, with a Pearson correlation coefficient of 0.91 ($p < 7e-243$). Note that the presence of vascular endothelial cell methylomes in the current atlas provides a qualitative difference, and vascular endothelial cell cfDNA was not possible in previous studies.

4) The study is performed on HG19, presumably because of the wealth of ChIP data that is available mapped to that build. However, newer builds are now available and it would be useful to consider using those instead.

We thank the reviewer. Indeed we have now re-mapped all the sequenced data to hg38 as well, and are uploading the processed files to GEO (bigwig, beta) for hg38, in addition to hg19.

5) It would be helpful if the authors provide a brief description of the formats in which the data is made available

Descriptions of the different formats are now included in the Data Availability section. We also included a more detailed description in the wgbstools github repository https://github.com/nloyfer/wgbs_tools.

Reviewer Reports on the First Revision:

Referees' comments:

Referee #1 (Remarks to the Author):

The authors would like to provide a resource and hence a core request was on getting more QC information on the data. This was only partially addressed and there seems to be quite some variation. Specifically:

The bisulfite conversion rate should be determined using the CC dinucleotide and not non-CG methylation. It should then be reported for each sample and ideally its should be above 99%. This was not done.

Non-CpG is not informative. Most of this is CpA methylation, which is well established and the information should be provided more accessible to the reader.

The quality of the data seems to vary a lot. For instance one Adipocyte sample shows 4.1% non-CpG methylation while the other two have 1.1%. This is a huge difference and unlikely true biology. Similarly one Kidney sample has 3.2% while the others all have 0.9%. This is true for many more.

Also for the CpG methylation, Colon fibroblasts have 73% CpG methylation while heart fibroblast have 60-65%...it is impossible to disentangle with the current information what is biology and what is data quality (conversion, library complexity, feature coverage distribution,...). More information needs to be provided.

Two samples have very low coverage 4-6x and only 4.9% of CpGs covered at 10x. Those can be removed as they will not be useful.

So in sum, the quality of the data remains a bit difficult to assess.

The second major point was that the authors did not provide much new biology. Reading the revised version that has not changed and the major display items also have not changed. While it is fine and very helpful for the community to have a resource paper, I would put the bar for a Nature paper a bit higher. The authors had the chance and time to dig a little deeper, but have not done that.

As a result the few but crucial points of this reviewer have not yet been addressed.

Referee #2 (Remarks to the Author):

The revised version of the MS is a significant improvement. The authors have done a great job addressing all my concerns. I only have 2 remaining concerns/comments:

1) The authors have added substantial data on the assessment of purity of their samples. Whilst this

demonstrates that purity is acceptable for a substantial number of samples, it should be observed that some samples are not that pure (e.g. endothelial cells in pancreas, adipocytes in breast or fibroblasts in GI-tract, all of these only display around 70% purity if not less, as shown in EDF1 and EDF2, respectively). I think that this minor limitation needs to be pointed out somewhere in the MS, and maybe the authors can find a way to “flag” the less pure samples in Table.S1 so that future users of this resource are aware of this.

2) Whilst one very important application of the author’s resource is to cfDNAm data, a major drawback of the resource remains in that it appears limited to DNAm data generated with WGBS or similar, since these technologies are, generally speaking, not used in the context of larger epigenome studies profiling 100s to 1000s of samples where cell-type deconvolution is also needed. These latter studies tend to use EPIC/450k arrays. Hence, for the task of performing cell-type deconvolution on the large numbers of solid tissue DNAm datasets generated with Illumina EPIC/450k arrays, this resource appears limited because of the very small overlap of EPIC probes with the ~1000 cell-type specific unmethylated marker regions (i.e. only around 10-20% of EPIC probes are present). In principle, because the regions identified by the authors display high spatial DNAm correlations, if the overlap with say EPIC probes was bigger, then this would not be a limitation, raising the question as to whether the authors should not have tried to also build cell-type specific marker regions that do overlap with say EPIC probes. The reader is thus left wondering whether there are indeed sufficient highly cell-type specific and high-confidence marker regions that do overlap EPIC probes. Clearly, by focusing on just the top-ranked 25 marker regions per cell-type, the overlap with EPIC probes would be very low, but the key question is whether there are more cell-type specific marker regions among the top-100 or top-250, that do overlap with EPIC probes, and which would then open up the possibility of using this resource for cell-type deconvolution of EPIC/450k datasets? As a benchmark would it not be a good idea to select blood cell-type specific marker regions (overlapping EPIC/450k probes) from their WGBS data, build a DNAm reference matrix for blood tissue, and then test this in an array context on in-silico mixtures of purified blood cell-types or on array datasets with matched FACS cell counts (such data is available in the public domain). My expectation is that the DNAm reference matrix built from their own WGBS data should work as well as those built with array data. Alternatively, if the authors were to use lower-ranked cell-type specific regions that do overlap with EPIC/450k probes, then they could apply their DNAm-atlas to the array-based data from Moss et al and cross-check predictions? Addressing these points would then open up potential applications to other tissue-types. Whilst I don’t consider these last requests to be a necessary revision, my feeling is that the MS would be stronger if it could address them.

Referee #3 (Remarks to the Author):

The authors have addressed all of my concerns

Author Rebuttals to First Revision:

Detailed response to reviewer comments

We thank the reviewers for their thoughtful comments and overall support of the manuscript. We have addressed all concerns, as detailed in the point by point rebuttal below, and have incorporated the new analyses and clarifications in the revised manuscript and the supplemental information. Most importantly, we have provided detailed information on the quality of samples, and generated new sets of markers that can be used with 450K/EPIC and various hybrid capture panels.

Referee #1 (Remarks to the Author):

The authors would like to provide a resource and hence a core request was on getting more QC information on the data. This was only partially addressed and there seems to be quite some variation. Specifically:

We thank the reviewer for noticing this. We have traced the variation in conversion rates to samples mapped to the human genome at two different computational clusters. We have now remapped all 205 samples (after removal of two cardiomyocyte samples that were sequenced at 6x), while properly clipping alignment mismatches at fragment ends, which were the major source of unconverted cytosines. As detailed in Table S1, the average CC conversion rate is now to 99.13%, with a median value of 99.2%. 201 samples showed CC conversion rate $\geq 98.8\%$, with the remaining four $\geq 98.5\%$.

This hardly affected CpG methylation levels. The average Pearson correlation between “old” and “new” methylation values was 0.995 (std=0.003; median=0.996). The median change in methylation levels was 0.9%, with an average of 1.17% of CpGs changing by 10% or more. We have now uploaded the revised bigwig and beta files to GEO, and updated all relevant figures and tables accordingly.

We also included a full set of QC measures, including:

- Read count (in pairs)
- Genome coverage (in read pairs)
- Percent of mapped reads
- Percent of proper pairs
- Duplication rate
- Mean fragment length
- CpG coverage (in read pairs)
- Percent of CpGs at $\geq 1x$ coverage
- Percent of CpGs at $\geq 10x$ coverage
- Mean Non-CpG methylation
- Mean CpA methylation levels
- Mean CC conversion rate
- Mean cytosine methylation (λ)
- Average CpG methylation

The bisulfite conversion rate should be determined using the CC dinucleotide and not non-CG methylation. It should then be reported for each sample and ideally its should be above 99%. This was not done.

CC conversion rates are now reported in Table S1 for all samples. The average conversion rate across all 205 samples is 99.13%, median rate is 99.2%, and all samples $\geq 98.5\%$.

Non-CpG is not informative. Most of this is CpA methylation, which is well established and the information should be provided more accessible to the reader.

CpA methylation (as well as non-CpG levels) are reported in Table S1 for all samples. Intriguingly, sorted neurons show elevated CpA methylation (near 10%) compared to 1% in other samples.

The quality of the data seems to vary a lot. For instance one Adipocyte sample shows 4.1% non-CpG methylation while the other two have 1.1%. This is a huge difference and unlikely true biology. Similarly one Kidney sample has 3.2% while the others all have 0.9%. This is true for many more.

We thank the reviewer. Indeed after remapping these changes disappear. All adipocyte samples show non-CpG methylation levels of 1.1-1.4% (CC conversion rates of 98.5%-98.9%). Similarly, all Kidney samples now show CC rates of 99.2% (Table S1).

Also for the CpG methylation, Colon fibroblasts have 73% CpG methylation while heart fibroblast have 60-65%...it is impossible to disentangle with the current information what is biology and what is data quality (conversion, library complexity, feature coverage distribution,...). More information needs to be provided.

Colon fibroblasts show 73-74% CpG methylation, which is indeed higher than cardiac fibroblasts (64%) or smooth muscle cells (65%), but very similar to dermal fibroblasts (73%), as well as other samples. These samples seem adequate according to all quality measures in Table S1. Intriguingly, the top differentially methylated regions between heart fibroblasts and cardiomyocytes include regions near 10 collagen genes, fibroblast growth factors, fibronectin, and more.

Two samples have very low coverage 4-6x and only 4.9% of CpGs covered at 10x. Those can be removed as they will not be useful.

These two Cardiomyocyte samples were initially kept based on their high similarity to the other four samples, which were properly sequenced. We have now removed them and reanalyzed the atlas based on 205 samples.

So in sum, the quality of the data remains a bit difficult to assess.

We hope that the updated Table S1 now highlights the overall quality of the data.

The second major point was that the authors did not provide much new biology. Reading the revised version that has not changed and the major display items also have not changed. While it is fine and very helpful for the community to have a resource paper, I would put the bar for a Nature paper a bit higher. The authors had the chance and time to dig a little deeper, but have not done that. As a result the few but crucial points of this reviewer have not yet been addressed.

This manuscript aims to serve as a resource for the community, and the results provided here contain highlights of this unique methylation atlas, including cell type-specific markers, biological interpretation of their annotations and regulatory function, a cell type-specific compendium of putative enhancers and unmethylated regions in each cell type, computational software suite for data analysis and deconvolution of cell-free DNA, and demonstration of possible applications. We hope that these could serve as a basis for further research by the community.

Referee #2 (Remarks to the Author):

The revised version of the MS is a significant improvement. The authors have done a great job addressing all my concerns. I only have 2 remaining concerns/comments:

1) The authors have added substantial data on the assessment of purity of their samples. Whilst this demonstrates that purity is acceptable for a substantial number of samples, it should be observed that some samples are not that pure (e.g. endothelial cells in pancreas, adipocytes in breast or fibroblasts in GI-tract, all of these only display around 70% purity if not less, as shown in EDF1 and EDF2, respectively). I think that this minor limitation needs to be pointed out somewhere in the MS, and maybe the authors can find a way to “flag” the less pure samples in Table.S1 so that future users of this resource are aware of this.

We thank the reviewer, and added a “purity estimate” column in Table S1. We also report the lower purity of several samples in the main text, including colon fibroblasts (78%), smooth muscle cells (82%), endothelial cells (86%), and adipocytes (87%).

2) Whilst one very important application of the author’s resource is to cfDNAm data, a major drawback of the resource remains in that it appears limited to DNAm data generated with WGBS or similar, since these technologies are, generally speaking, not used in the context of larger epigenome studies profiling 100s to 1000s of samples where cell-type deconvolution is also needed. These latter studies tend to use EPIC/450k arrays. Hence, for the task of performing cell-type deconvolution on the large numbers of solid tissue DNAm datasets generated with Illumina EPIC/450k arrays, this resource appears limited because of the very small overlap of EPIC probes with the ~1000 cell-type specific unmethylated marker regions (i.e. only around 10-20% of EPIC probes are present). In principle, because the regions identified by the authors display high spatial DNAm correlations, if the overlap with say EPIC probes was bigger, then this would not be a limitation, raising the question as to whether the authors should not have tried to also build cell-type specific marker regions that do overlap with say EPIC probes. The reader is thus left wondering whether there are indeed sufficient highly cell-type specific and high-confidence marker regions that do overlap EPIC probes. Clearly, by focusing on just the top-ranked 25 marker regions per cell-type, the overlap with EPIC probes would be very low, but the key question is whether there are more cell-type specific marker regions among the top-100 or top-250, that do overlap with EPIC probes, and which would then open up the possibility of using this resource for cell-type deconvolution of EPIC/450k datasets? As a benchmark would it not be a good idea to select blood cell-type specific marker regions (overlapping EPIC/450k probes) from their WGBS data, build a DNAm reference matrix for blood tissue, and then test

this in an array context on in-silico mixtures of purified blood cell-types or on array datasets with matched FACS cell counts (such data is available in the public domain). My expectation is that the DNAm reference matrix built from their own WGBS data should work as well as those built with array data. Alternatively, if the authors were to use lower-ranked cell-type specific regions that do overlap with EPIC/450k probes, then they could apply their DNAm-atlas to the array-based data from Moss et al and cross-check predictions? Addressing these points would then open up potential applications to other tissue-types. Whilst I don't consider these last requests to be a necessary revision, my feeling is that the MS would be stronger if it could address them.

We thank the reviewer. Indeed, there are sufficient cell type-specific marker regions that directly overlap 450K or EPIC CpG sites. Extended Table S12 and S13 now list the top 25 unmethylated markers for each cell type, that overlap Illumina 450K and EPIC BeadChip arrays, respectively. These tables list the most cell type-specific differentially unmethylated probes for each cell type (936 CpG sites for 450K and 950 sites for EPIC), as well as their surrounding DMR regions, which could be useful for custom hybrid capture panels that target these methylation arrays regions. We hope that these markers could be integrated into array-based panels for a more accurate analysis of DNA methylation array data. The new Extended Figure S8A-B now depict these genomic regions:

Nonetheless, further analysis of such array data is outside the scope of this manuscript, and some normalizations might be required, similarly to normalization methods that are used for integration of type 1 and type 2 DNA methylation probes.

In addition to these new atlases, we have also generated similar atlases for RRBS regions (Extended Table S14, 934 regions) and for commercial hybrid capture DNA methylation panels, including Illumina's TruSeq EPIC (S15, 946 regions), Roche's SeqCapEpi (S16, 944 regions), and Agilent's SureSelectXT panel (S17, 947 regions).

We hope that these markers will allow researchers to analyze various datasets based on methylation arrays and sequencing, in addition to whole-genome DNA methylation sequencing data.

Referee #3 (Remarks to the Author):

The authors have addressed all of my concerns

Reviewer Reports on the Second Revision:

Referee #2 (Remarks to the Author):

Overall, I am enthusiastic about this resource, but it is a great pity that the authors are unable to demonstrate use and added value of their DNAm-atlas on Illumina-array data or other technologies (e.g. RRBS), since all the existing largest sample-size DNAm studies have not actually used WGBS.

Author Rebuttals to Second Revision:

Detailed response to reviewer comments

Reviewer 2 had two remaining concerns, as summarized by the editor. He/she wanted to see additional evidence for the **broad applicability of the atlas to analyzing previous array data**, and also requested that the **limitations of purity of cell preparations** used to generate the atlas, be acknowledged and discussed.

We thank the reviewer for their support and constructive comments. We have now **applied our WGBS markers to additional 1,770 DNAm arrays**. We constructed a specialized atlas overlapping 450K CpGs (Table S12, presented in the previous version), and applied it to previously published methylation array data. As described below, the key utility of our atlas for interpreting array data is the generation of methylomes of primary cell types that were not analyzed before, such as pancreatic alpha and delta cells, lung alveolar and bronchial epithelial cells and more. Furthermore, the exhaustive atlas allows to identify novel, high quality markers of other cell types that were profiled before.

The new data, described in Extended Figure S9, include 87 BeadChip 450K arrays measuring the methylome of pancreatic islets (Extended Figure S9A); 865 arrays from TCGA portraying normal and cancerous lung methylomes (Extended Figure S9B); and 818 arrays characterizing the methylomes of breast biopsies from TCGA (Extended Figure S9C).

Overall, the new analysis reveals the relative proportions of cell types that could not have been identified before. In the islet methylomes, we can now determine the proportion of alpha and delta cells (note that islet cell composition depends on the procedure of isolation, and is an important factor for biological analysis and clinical utility - hence the difference between Figure 6I and Extended Figure S9); in lung methylomes, we show that alveolar cell DNA dominates lung adenocarcinomas while bronchial cell DNA is present in small cell lung cancer and squamous cell carcinomas; in breast methylomes, we show that basal and luminal cell methylation markers are indicative of breast cancer subtypes as determined by PAM50 gene expression analysis. Thus, the new atlas has broad applicability not only to WGBS, but also to the widely used methylation array data. The new Extended Figure S9 is pasted below, and is referred to from the main text (Discussion).

Extended Figure S9. Deconvolution of previously published 450K DNA methylation array data. (A) Deconvolution of pancreatic islet methylomes. Methylation arrays from 53 male and 34 female non-diabetic donors⁴⁹ were analyzed, revealing detailed cellular composition including previously uncharacterized alpha and delta cells. No statistically significant sex differences in cellular composition were observed. **(B) Analysis of 865 pulmonary methylomes from TCGA⁵⁰.** WGBS-based markers for lung alveolar epithelium and lung bronchial epithelium cells reveal differential cell populations in 443 LUAD, 11 SCLC, 337 LUSC, 32 Normal adjacent (LUAD), and 42 Normal adjacent (LUSC) lung methylomes. Note that alveolar cell DNA dominates lung adenocarcinomas, while bronchial cell DNA is present in small cell lung cancer and squamous cell carcinomas. **(C) DNA methylation from 721 cancerous and 97 normal breast biopsies from TCGA.** WGBS-based markers for breast luminal and basal epithelial cells were used to study the cellular composition in TCGA⁵¹, which were classified into five subtypes using PAM50, a 50-gene expression-based classification⁶⁵. Different cell composition is observed for Normal-like, Basal-like, Luminal A, Luminal B, and Her2-enriched PAM50 subtypes, compared to healthy breast biopsies.

We also revised the manuscript to further discuss the purity of the atlas, including possible limitations of FACS sorting. Additionally, the estimated purity of each cell type is listed in the main data table (Supplemental Table S1), including sorting comments, and the FACS gating plots used for each cell type are detailed in the “Supplementary Information” file. The revised text discusses purity in the Results section as well as the Discussion.

Results p.4: “Few samples show lower purity (e.g. colon fibroblasts 78%, smooth muscle cells 82%, endothelial cells 86%, or adipocytes 87%).”

Results p.16: “Importantly, a similar deconvolution of the 205 samples presented here, yields an average contribution of 94% for the expected cell type for each sample (median of 95%, Extended Table S10), or 91% (median of 92%) in a more stringent

leave-one-out cross validation analysis (Extended Table S11), highlighting the purity of collected samples.”

Discussion p.18: “We also acknowledge that the purity of the sorted cell populations varies, due to different quality of antibodies used for FACS sorting and the extent to which they allow separation of cell types. Nonetheless, even the least pure cell types in the atlas (e.g. some preparations of vascular endothelial cells, fibroblasts, smooth muscle cells and adipocytes that show 70-80% purity), when averaged over replicates, are useful for identification of differentially methylated regions and for inference of cell composition in mixtures.”